# Denitrification is a community trait with partial pathways dominating across microbial genomes and biomes

Grace Pold [1], Aurélien Saghaï [2], Christopher M. Jones[2] & Sara Hallin [2] ✉

Diverse microorganisms can execute one or more steps in denitrification, during which nitrate or nitrite is successively reduced into nitric oxide, nitrous oxide, and ultimately dinitrogen. Many of the best-characterized denitrifiers are complete denitrifiers capable of executing all steps in the pathway, but their dominance in natural communities and what metabolic traits and environmental factors drive the global distribution of complete vs. partial denitrifiers are unclear. To address this, we conducted a comparative analysis of denitrification genes in 61,293 genomes, 3991 metagenomes, and 413 terrestrial and aquatic metatranscriptomes. We show that partial denitrifiers outnumber complete denitrifiers and the potential to initiate denitrification is more common than the potential to terminate it, particularly in nutrient rich environments. Our results further indicate that complete denitrifiers tend to be fast-growing organisms, favoring organic acid over sugar metabolism, and encoding the ability to oxidize and reduce a broader range of organic and inorganic compounds compared to partial denitrifiers. This suggests complete denitrifiers are metabolically flexible opportunists. Together, our results indicate an environmental footprint on the presence of denitrification genes which favors the genomic potential for partial over complete denitrification in most biomes and highlight that completion of the denitrification pathway is a community effort.

The potent greenhouse gas and ozone depleting agent nitrous oxide ($N_2O$) is produced in several microbial N-cycle processes, of which denitrification plays a dual role in both producing and consuming $N_2O$. In this facultative anaerobic microbial respiratory pathway, nitrate ($NO_3^-$) or nitrite ($NO_2^-$) are used as electron acceptors and successively reduced into nitric oxide (NO), $N_2O$, and dinitrogen ($N_2$) by a diverse range of predominantly bacterial species. As the genes encoding the enzymes involved in these reductive steps can be independently gained and lost[1–3], they are found in different combinations among genomes[4], such that denitrification can be executed by complete denitrifiers with genes encoding enzymes for all steps or, alternatively, split between multiple partial denitrifier community members. Recent

studies suggest that genetically complete denitrifiers are scarce among genomes assembled from environmental data[5–10]. However, much of our understanding of the physiology and regulation of denitrification is based on studies of a very phylogenetically restricted set of complete denitrifiers[11–15]. Thus, there is pressing need for a broader look at the prevalence, diversity and ecology of organisms genetically capable of complete and partial denitrification, particularly considering their differential roles in serving as sources and sinks of $N_2O$[16].

Split pathways divide the protein production costs for a pathway between multiple cells and may enable higher ATP flux[17,18], increasing the specific growth rate. Although this could be a fitness advantage under nitrite-rich conditions, ATP yields per molecule nitrate or nitrite

[1]Department of Soil and Environment, Swedish University of Agricultural Sciences, Uppsala, Sweden. [2]Department of Forest Mycology and Plant Pathology, Swedish University of Agricultural Sciences, Uppsala, Sweden. ✉e-mail: sara.hallin@slu.se

would be lower for partial compared to complete denitrification. This is expected to lead to partial denitrification being favored when electron acceptor supply is not limiting, but complete denitrification being favored when the flux of electron donors such as carbon (C compounds is high relative to electron acceptor availability). Splitting the denitrification pathway also increases dependencies between community members, not only because the denitrification product of one organism serves as the electron acceptor for another[19,20], but also because the regulation of specific reactions may rely on intermediates that those executing the reaction themselves cannot use[21]. Further, there is less buildup of toxic intermediate compounds, which may allow microorganisms only capable of carrying out the initial steps of denitrification avoid the cytotoxic effects of $NO_2^-$, particularly under low pH environments[11,22]. This jointly indicates that split pathways may open more niches, thereby supporting a greater diversity of denitrifiers[20]. If organisms encoding partial vs. complete denitrification fulfill different niches, this should also be reflected in additional traits, such as growth rate, abiotic optima, and the ability to metabolize different substrates and use electron acceptors beyond those in the denitrification pathway. For instance, if complete denitrification is indeed associated with C influx, complete denitrifiers would be expected to be capable of higher growth and broader substrate catabolism compared to partial denitrifiers. Addressing the linkages between the complete or partial denitrification and the capacity for other traits can help identify the broader roles and conditions that favor each denitrifier type.

Here we combined comparative genomics (Fig. 1a) with a global survey of denitrification gene ratios in environmental metagenomes and metatranscriptomes to evaluate the diversity of bacterial denitrifier types, their metabolic traits, and their distribution across

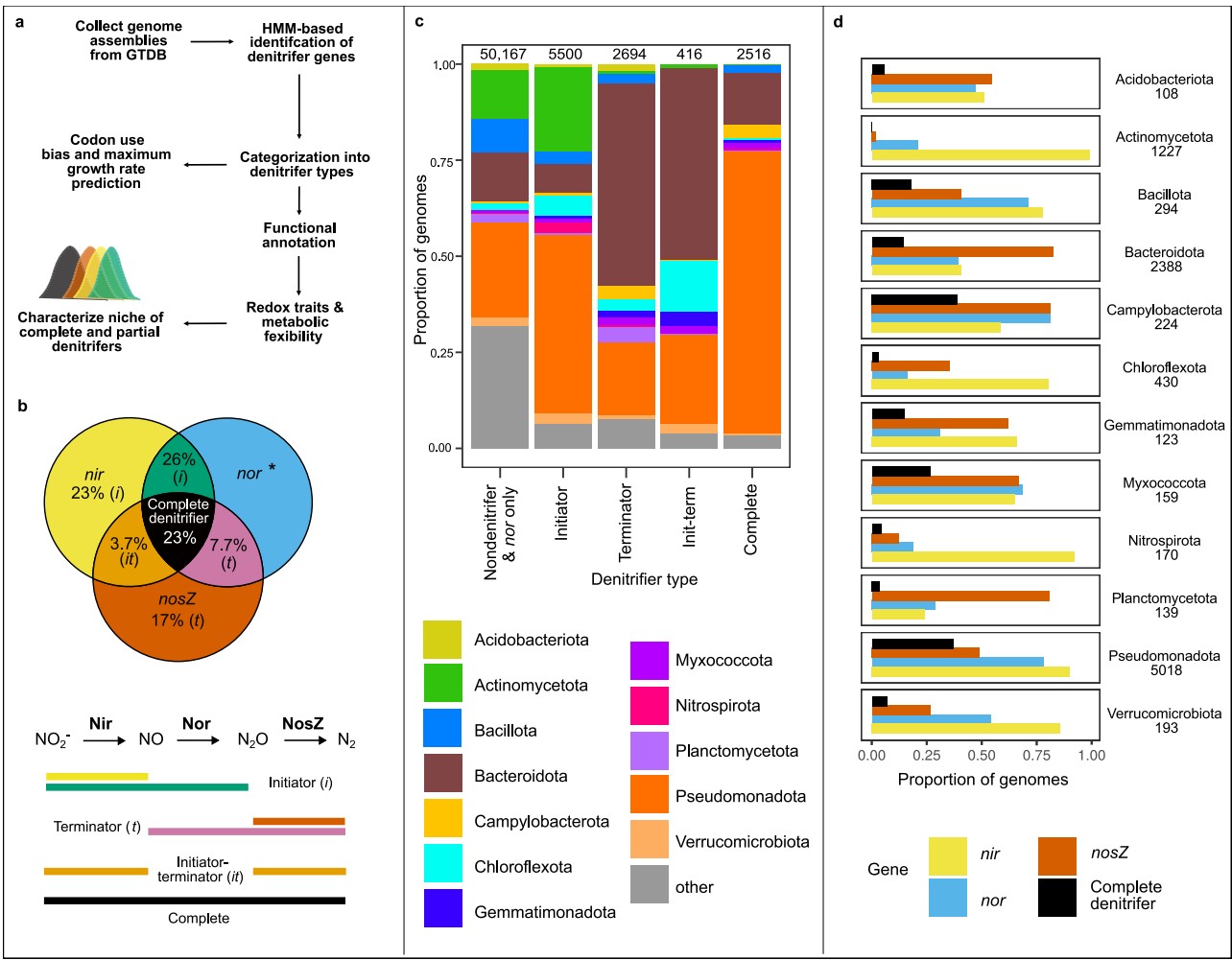

**Fig. 1 | Methods overview and phylogenetic distribution of denitrifier types.**
**a** Overview of experimental approach for comparative genomics. Metagenome-assembled, single cell and isolate bacterial genomes covering environmental, host-associated, and engineered biomes were extracted from GTDB v.214.1 (Supplementary Data 2). Denitrification genes were annotated using a combination of HMM-based searches, phylogeny construction, and manual inspection of alignments. Denitrifier genomes were then searched for additional functional traits including maximum growth rate, transcription factors and potential to use various electron donors and acceptors. **b** Venn diagram of prevalence of denitrifier types. The number inside each circle denotes the percentage of denitrifier genomes with that genotype. Letters in italics denote whether genotype is considered an initiator (*i*), terminator (*t*) or is capable of both (init-term; *it*). Reactions completed by each enzyme and corresponding denitrifier type name are listed below the diagram, with the lines under each step defining the colors used in the Venn diagram. Genomes encoding *nor* without accompanying *nir* or *nosZ* genes are marked with an asterisk and classified as non-denitrifiers. **c** Stacked bar charts denoting relative proportion of genomes belonging to phyla within each denitrifier type, with non-denitrifiers defined as containing neither *nir* nor *nosZ*, or being archaeal nitrifiers or anammox bacteria (see "Methods"). The number of genomes corresponding to each denitrifier type is indicated above the corresponding bar. Phyla represented by fewer than 100 denitrifier genomes are aggregated under "other" and can be found in Supplementary Data 11. **d** Proportion of denitrifier genomes within dominant phyla encoding each denitrification enzyme gene as well as the complete denitrification pathway. Numbers to the right in (**d**) indicate the number of denitrifier genomes belonging to each phylum denoted in the graph. Source data are provided as a Source Data file.

engineered, environmental and host-associated biomes. We posited that environmental factors favoring or disfavoring execution of the different steps leave a genetic footprint in the form of presence and absence of genes encoding the enzymes responsible for them, as well as via differential associations between denitrification enzyme genes and redox traits. Furthermore, considering organisms encoding a complete denitrification pathway are able to use a broader range of nitrogenous terminal electron acceptors than those encoding fewer, we hypothesized that the former would have broader metabolic flexibility. We used a framework in which partial denitrifiers include genetically defined initiators, which encode genes involved in $NO_2^-$ but not $N_2O$ reduction, and terminators, which encode genes for $N_2O$ but not $NO_2^-$ reduction (Fig. 1b). We used $NO_2^-$ reduction, the decisive step of denitrification[23], as the first point of our analysis. We combined comparative genomics of 61,293 non-redundant bacterial isolate, metagenome assembled (MAGs), and single cell genomes with 3991 metagenomes derived from all major biomes and 413 terrestrial and aquatic metatranscriptomes to determine the frequency of a complete vs. partial denitrification pathway and the genetic potential to initiate vs. terminate denitrification at the genome and community level. Here we show evidence for dominance of partial over complete denitrifiers in genomes, metagenomes, and metatranscriptomes. We interpret our results in the context of resource availability and complementary genomic traits associated with the denitrifier types in our framework and propose that complete denitrifiers are metabolically flexible generalists.

## Results

### Partial denitrification dominates microbial genomes

We assessed the prevalence of partial and complete denitrification pathways among 61,293 bacterial genomes represented in GTDB release 214.1 (Fig. 1a). Genes for the denitrification enzymes NirK (*nirK*) and NirS (*nirS*) catalyzing $NO_2^-$ reduction to NO, all proposed NO-reducing heme copper oxidases (i.e., Nor (*nor*)[24]), and NosZ (*nosZ*I, *nosZ*II) catalyzing $N_2O$ reduction were annotated using hidden Markov models trained on manually curated databases of these genes[25]. Genomes carrying *nor* but not *nir* or *nosZ* accounted for 5% of the total number of assemblies and were excluded in downstream analyses because Nor in *nor*-only genomes are likely involved in detoxification rather than respiration. With this conservative approach, 18% of the genomes (n = 11,126) were considered denitrifiers and included *nir* and/or *nosZ* with or without *nor*. Among these, 23% encoded genes for complete $NO_2^-$ reduction to $N_2$ (complete denitrifiers; Fig. 1b), with the complete denitrification trait more prevalent among isolates compared to MAGs (29% vs. 17%, respectively, Supplementary Table 1). However, there was considerable variation between phyla (Fig. 1c, Supplementary Fig. 1). Twenty-eight different bacterial phyla encoded the complete denitrification trait, with an overwhelming dominance of Pseudomonadota in our dataset (Fig. 1c). Among phyla represented by at least 100 denitrifier genomes, complete denitrifiers were most prevalent within Campylobacterota (38% of denitrifiers), Pseudomonadota (37%), and Myxococcota (26%, Fig. 1d).

Using our framework of initiators and terminators, we found that initiators encoding *nir* with or without *nor* but not *nosZ* were broadly more prevalent than terminators having *nosZ* with or without *nor* but not *nir* (49% vs. 24% of genomes with denitrification genes, respectively). This may represent the potentially divergent functions for $NO_2^-$ versus $N_2O$ reduction as a means of growth or detoxification vs. maintenance[26]. Partial denitrifiers occurred in 68 bacterial phyla, with initiators being overrepresented within Actinomycetota (98% initiators, 1% terminators) and Nitrospirota (88%, 8%) (Fig. 1d). However, Nitrospirota accounted for just a small fraction of initiator genomes (2.7%), while the majority were Pseudomonadota (47%) and Actinomycetota (22%) (Fig. 1c). We found a higher proportion of terminators than initiators within Planctomycetota (76% terminators, 19%

initiators) and Bacteroidota (60%, 18%) (Fig. 1d), with the latter phylum accounting for the majority of terminators in the dataset (53%; Fig. 1c). We also found genomes carrying *nir* and *nosZ* but not *nor*, suggesting a third category: the initiator-terminators. They accounted for just 4% of genomes with denitrification genes, including 13% of Gemmatimonadota, 13% of Chloroflexota, 9% of Bacteroidota and 5% of Verrucomicrobiota (Fig. 1c). It is possible that these organisms either depend on NO reduction by other organisms or produce $N_2O$ via pathways that do not depend on Nor[27–30].

Among all genomes carrying *nosZ* (i.e., terminators, initiator-terminators, and complete denitrifiers), those encoding clade I *nosZ* were more likely to encode the complete denitrification trait compared to genomes encoding clade II (74% in clade I vs. 26% in clade II; Supplementary Fig. 2), as previously observed[4]. Within clade II, complete denitrification was most prevalent in Pseudomonadota and Aquificota, with 80% of Pseudomonadota and 94% of Aquificota encoding complete denitrification compared to an average of 20% in the remaining phyla. This suggests that the difference in clade I vs. clade II prevalence can be used as a proxy for complete vs. partial denitrifier prevalence within the $N_2O$ reducing community, but not if clade II is dominated by Pseudomonadota or Aquificota.

### Inferred ecological preferences of complete vs. partial denitrifiers

We next assessed whether the genetic potential for complete denitrification or for initiating rather than terminating denitrification was associated with the predicted growth rate and resource use patterns among bacteria. Using the codon bias-based maximum growth rate prediction tool gRodon[31], we categorized genomes into fast- (growth rate faster than $0.2\,h^{-1}$) and slow-growing (growth rate lower than $0.2\,h^{-1}$)[31]. Rapid growth was more common among complete denitrifiers (83%) compared to initiators (70%), terminators (65%) and initiator-terminators (62%; Fig. 2a upper panel) and for genomes encoding clade I (85%) compared to clade II NosZ (65%; Fig. 2a lower panel). Similarly, estimated median growth rates of complete denitrifiers ($0.46\,h^{-1}$) were greater than that of initiators ($0.28\,h^{-1}$), terminators ($0.23\,h^{-1}$) and initiator-terminators ($0.20\,h^{-1}$; overall Kruskall-Wallis $\chi^2 = 401$, Dunn $P < 0.0001$ in all cases). This faster potential growth rate was associated with a lower sugar to acid preference of complete denitrifiers compared to all three of the categories of partial denitrifiers, based on the total sum of genes involved in each pathway[32] (Fig. 2b). This was accompanied by a higher density of transcription factors (Fig. 2c) and transporters (68 Mbp$^{-1}$ in complete denitrifiers vs. 65 Mbp$^{-1}$ in initiators, 40 Mbp$^{-1}$ in terminators, and 42 Mbp$^{-1}$ in initiator-terminators, $P < 2 \times 10^{-16}$). Furthermore, complete denitrifiers were inferred to grow on more of the 56 organic substrates examined (median of 20 corresponding to 4.75 Mbp$^{-1}$), compared to initiators (13; 3.32 Mbp$^{-1}$), terminators (11; 2.90 Mbp$^{-1}$), and initiator-terminators (9; 2.55 Mbp$^{-1}$; Fig. 2d). This was driven by complete denitrifiers using a broader range of organic and amino acids compared to partial denitrifiers (median of 15 vs. 6–7), rather than sugars (median of 2 vs. 2–3).

We subsequently examined the potential for partial and complete denitrifiers to use a range of inorganic compounds as electron donors and acceptors (Fig. 2e). While the ability to oxidize inorganic compounds such as hydrogen may be associated with an ability to tolerate energy-limited conditions[33], the ability to reduce inorganic compounds may indicate that one type or another of denitrifiers is better able to tolerate lower redox conditions. Although most genomes lacked the genetic potential to oxidize inorganic substances or reduce non-nitrogenous terminal electron acceptors (median = 0) other than oxygen, complete denitrifiers had the genetic potential to both oxidize and reduce more compounds than partial denitrifiers (Kolmogorov-Smirnov statistic $P < 0.0001$ in all cases). Almost all (99%) genomes encoded the potential for aerobic respiration, with low affinity oxidases more prevalent in complete denitrifiers (87%) than initiators

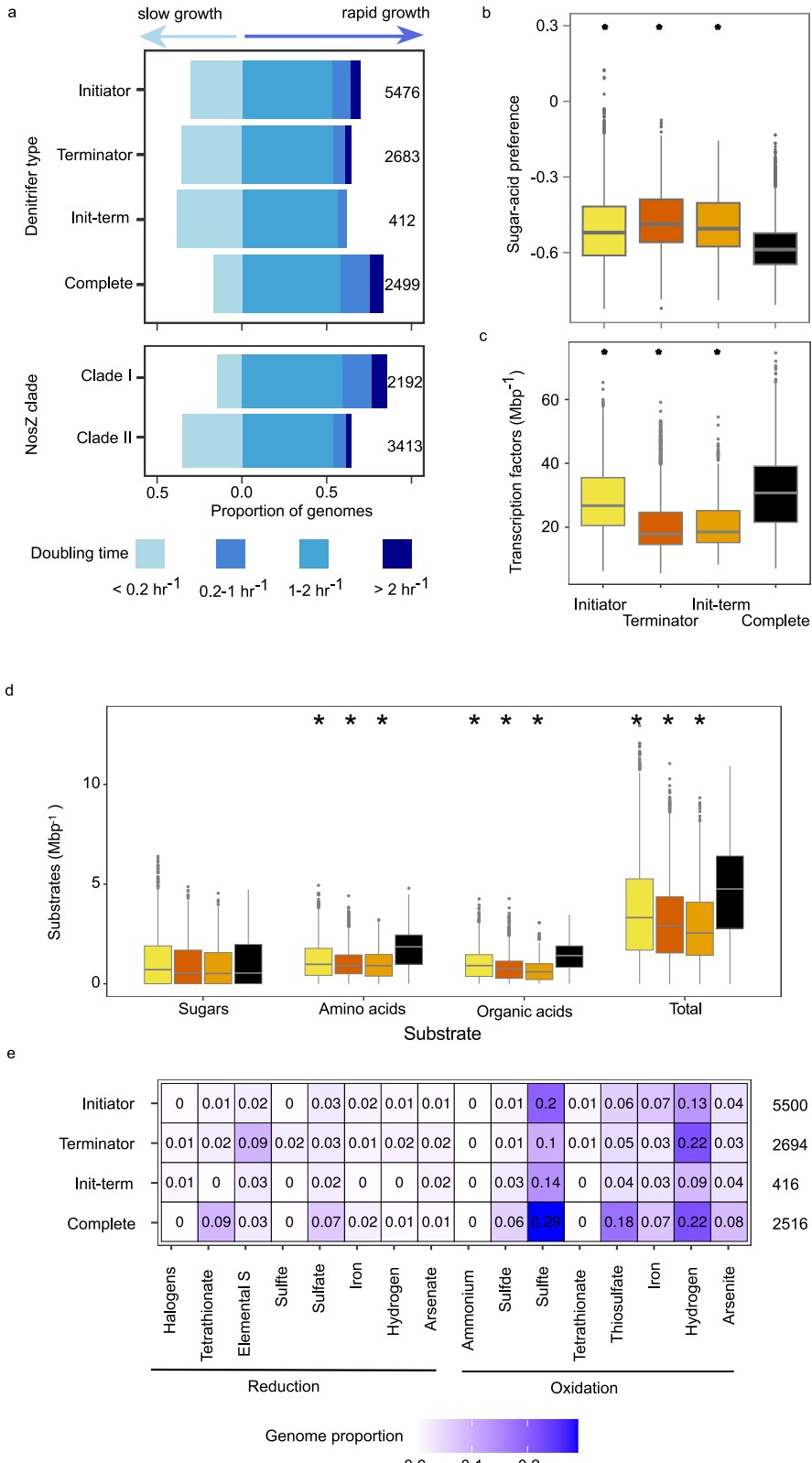

(83%; $\chi^2 = 15.03$, $P = 0.0001$), and similar in prevalence to terminators (85%; $\chi^2 = 0.083$, $P = 0.7734$) and initiator-terminators (86%; $\chi^2 = 0.91$, $P = 0.3397$). High affinity variants were more prevalent among complete denitrifiers (99%) vs. initiators (89%; $\chi^2 = 228$, $P < 2.2 \times 10^{-16}$), terminators (94%; $\chi^2 = 84.63$, $P < 2.2 \times 10^{-16}$) and initiator-terminators (88%; $\chi^2 = 142.82$, $P < 2.2 \times 10^{-16}$).

The inferred abiotic niche breadths of complete denitrifiers were not uniformly broader than that of partial denitrifiers. Complete denitrifiers had slightly broader inferred temperature ranges compared to initiators (26.84 vs. 25.12 °C; $t = -24.583$, $P < 2 \times 10^{-16}$), terminators (25.54 °C, $t = -18.569$, $P < 2 \times 10^{-16}$) and initiator-terminators (25.93 °C, $t = -8.038$, $P = 1.02 \times 10^{-15}$). pH ranges were narrower in

**Fig. 2 | Genome-inferred traits of complete and partial denitrifier bacteria.**
**a** Estimated maximum growth rate of organisms based on codon usage bias.
Stacked bar charts show the maximum predicted growth rate for genomes separated by denitrifier type (upper panel) and NosZ clade (lower panel). Slow-growing taxa (maximum growth rate <0.2 h$^{-1}$) are depicted to the left on the horizontal axis and fast-growing taxa to the right. Numbers to the right denote number of genomes used in maximum growth rate predictions and exclude genomes with fewer than 10 ribosomal proteins annotated. **b** Boxplots of sugar and organic acid preference of denitrifiers, based on KEGG annotation ratios[32]. **c** Genomic density of transcription factors inferred using DeepTFactor. **d** Boxplot of substrate use counts by denitrifier type and substrate category based on GapMind, with colors following (**b**, **c**).

**e** Heatmap of terminal electron acceptors and donors, with intensity of color and number denoting proportion of genomes within denitrifier type encoding redox ability. Terminal oxidases are near-ubiquitous and are not included in figure or totals. In (**b**–**d**), box boundaries represent first and third quartiles, with midline denoting median. Whiskers denote the 1.5 IQR, and outliers are shown as individual points. Asterisks above boxes indicate a difference between complete denitrifiers and the partial denitrifier type and no comparisons between partial denitrifier types were made. Numbers to the right in (**e**) denote number of genomes used in (**b**–**e**). Elemental S elemental sulfur, Init-term initiator-terminator. Source data are provided as a Source Data file.

complete denitrifiers compared to initiators (3.75 vs. 3.83; t = 8.136, $P = 4.44 \times 10^{-16}$) and broader compared to terminators (3.58; t = −6.756, $P = 1.50 \times 10^{-11}$). Salinity ranges were broader in complete denitrifiers than initiators (5.64 vs. 5.31; t = −3.775, $P = 0.000162$) but narrower than in initiator-terminators (5.92; t = −2.441, $P = 0.014665$).

We subsequently assessed the degree to which complete and partial denitrifiers were associated with high or low resource availability using CoverM[34] to map reads from the TARA Oceans[35] project to MAGs derived from the same data[36]. No subset of the metagenome data we collected or other databases we screened had a sufficient number of complete denitrifier MAGs and environmental data to allow for this analysis, highlighting the rarity of complete denitrifiers in global ecosystems. Nitrate was used as a proxy for denitrification electron acceptor availability and chlorophyll concentration as a proxy for electron donor supply because of its positive association with gross primary productivity[37]. The proportion of reads mapping to the collection of 170 MAGs in the dataset differed by sample and ranged from 5 to 30%. We calculated a standardized environmental response by correlating the proportion of reads mapping the denitrifier types among the MAGs (7 complete, 13 initiators, 14 terminators, 134 non-denitrifiers and 2 initiator-terminators) with chlorophyll and $NO_3^-$ concentrations in different samples and standardizing this to the maximum observed relative abundance of each MAG. We found that terminators and complete denitrifiers tended to have a greater increase in relative abundance with increasing $NO_3^-$ than non-denitrifiers, but there was no difference between complete denitrifiers and partial denitrifiers (Supplementary Fig. 3a). Mean standardized chlorophyll and chlorophyll: $NO_3^-$ ratio responses were similar across all groups of organisms, independent of denitrifier type (Supplementary Fig. 3b, c). One possible reason for the apparent lack of correlation between denitrifier type and resource availability is that these bulk measurements likely do not represent the microhabitat that denitrifiers experience. For instance, pre-filtering of the samples depleted much of the particulate organic matter, where denitrification activity is concentrated[38], and where modeling indicates that complete denitrifiers are favored[20]. Nonetheless, the results from this pelagic marine dataset do not indicate that measured resource availability affects the prevalence of complete or partial denitrifier genomes.

## Prevalence of denitrification initiators and terminators varies across global biomes

Next, we assessed variation in denitrification initiator and terminator prevalence and complete and partial denitrifiers among terminators (i.e., $N_2O$ reducers) at the community level in a broad range of environments. We screened 3991 metagenomes derived from soil, aquatic, engineered and host-associated biomes for *nirK*, *nirS*, and *nosZ* clades I and II using an HMM-based search and phylogenetic placement approach[39]. The difference in the copy number of *nosZ* and *nir* genes normalized to Gbp sequenced in each metagenome (δ*nos*-*nir*) was used as a proxy for the genetic potential for termination compared to initiation of denitrification, and *nosZ* clade I vs. clade II (δ*nosZ*I-*nosZ*II) as a proxy for complete denitrifier dominance within the $N_2O$ reducing community. Delta values were chosen for their ability to account for

differences in overall denitrifier prevalence between biomes, including cases where one or both of the denitrification genes being compared were not detected.

Across the majority of biomes, *nir* gene fragments were typically more abundant than *nosZ*, indicating a greater or similar potential to initiate than terminate denitrification within the denitrifier communities (Fig. 3a). Marine mats served as an exception, and *nosZ* prevalence exceeded that of *nir* by 60%. This would occur if there were higher rates of $NO_3^-$ assimilation or low inputs of $NO_2^-$ from nitrification[40], as suggested by the overall low prevalence of *nir* and lack of ammonifier associated *nirK* clades observed previously in these samples[25]. *nosZ* Clade II was more prevalent than *nosZ* Clade I in nearly all biomes considered, (Fig. 3b). This indicates that partial denitrifiers generally dominate terminator communities because partial denitrification is more prevalent among Clade II *nosZ*. This is further supported by our observation that phyla depleted in complete denitrifiers, such as Bacteroidetes, Chloroflexota and Gemmatimonadota, dominated *nosZ* clade II across biomes (Fig. 3c, d). Therefore, we can conclude that the majority of organisms capable of terminating denitrification in global biomes do not also initiate this process. This is particularly the case in croplands, marshes, and activated sludge from municipal wastewater treatment plants, which were the biomes where δ*nosZ*I-*nosZ*II is lowest for terrestrial, marine, and engineered environments (Fig. 3b). The dominance of *nosZ*II coincides with the highest total *nosZ* and *nir* gene abundances (Supplementary Data 1), indicating conditions that are overall more favorable for denitrification also promote $N_2O$ reducers that are partial rather than complete denitrifiers. A notable exception to this pattern is sewage communities, which were dominated by clade I *nosZ* despite having high overall *nir* gene fragment prevalence, and seems to largely reflect the preponderance of *Acidovorax* in these samples[41].

We also assessed the correlation between the diversity of denitrifiers based on phylogenetic diversity of the dominant nitrite reductase gene, *nirK* and potential initiator-terminator or clade I and II differential abundance[25]. We found that *nirK* phylogenetic diversity was positively correlated with δ*nos*-*nir* in soils (Spearman's ρ = 0.46, $P < 2.2 \times 10^{-16}$) and marine samples (ρ = 0.19, $P = 1.18 \times 10^{-11}$), while *nirK* diversity was positively correlated with δ*nosZ*I-*nosZ*II in soils (ρ = 0.29, $P < 2.2 \times 10^{-16}$) and uncorrelated in marine samples (ρ = 0.03, $P = 0.35$). These results indicate that more diverse denitrifier communities occur where capacities for initiating and terminating denitrification are relatively more balanced, such that complete denitrification at the community level is associated with greater niche partitioning.

Environmental drivers of *nir* vs. *nosZ* and *nosZ* clade prevalence were assessed using random forest modeling on the largest soil and marine datasets with complete metadata represented in our analysis[42]. We generated accumulated local effect plots to show the main effect of the target variable on predicted δ*nos*-*nir* (Fig. 4) and δ*nosZ*I-*nosZ*II (Fig. 5), while accounting for the other predictors. Models explained less than half of the variance in gene differences, except for δ*nosZ*I-*nosZ*II in soils. Increases in soil organic carbon content up to ~1% positively affected δ*nos*-*nir*, while the effect of soil $NO_3^-$ content was always negative (Fig. 4a). *nosZ* was also predicted to become less

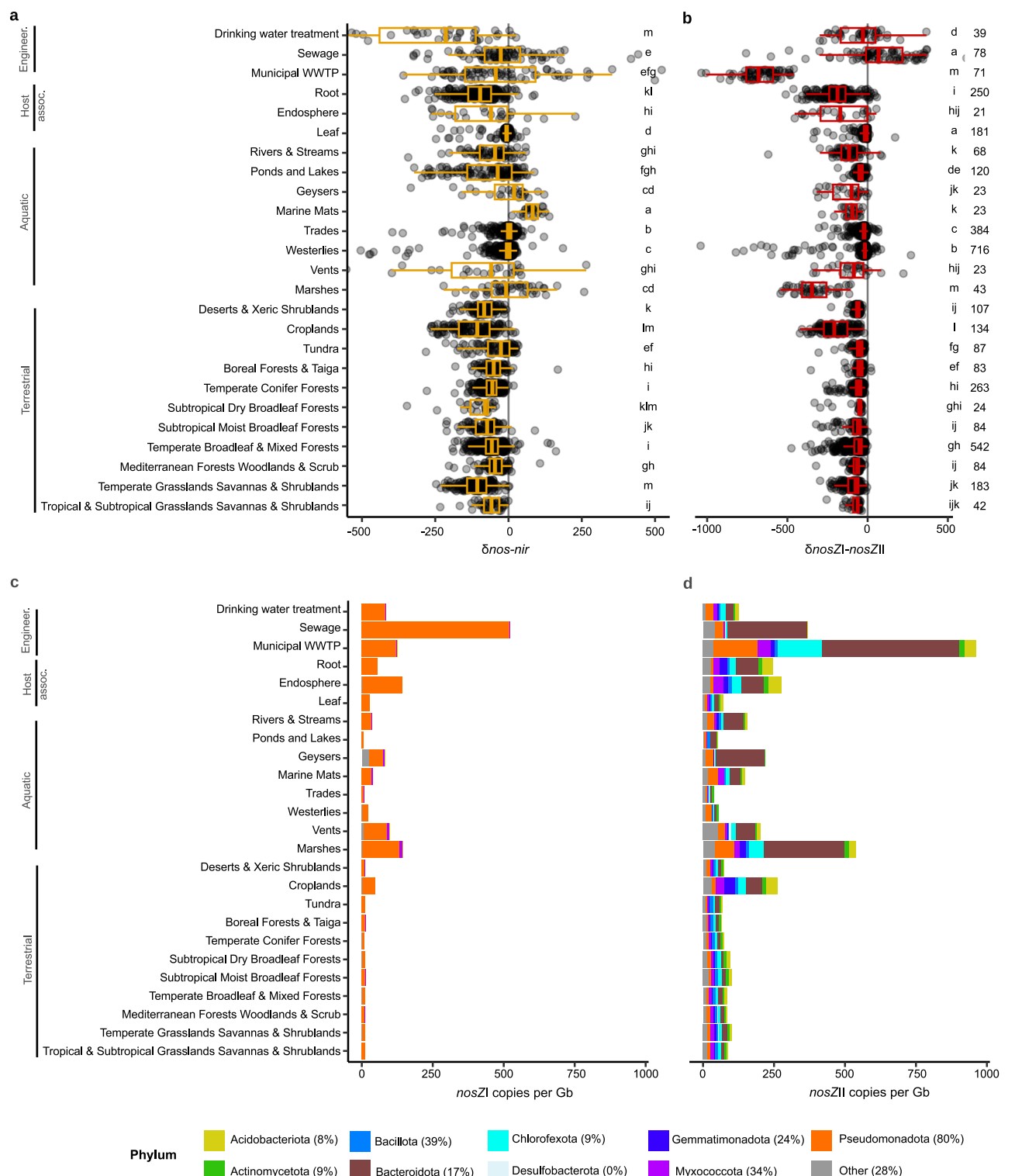

**Fig. 3 | Balance in prevalence of denitrification genes shows dominance of initiators at the community level across global biomes.** Boxplots showing difference in *nos* versus *nir* counts (($nosZ$ I + $nosZ$ II) − ($nirK$ + $nirS$)) (**a**); and clade I and clade II *nosZ* counts (**b**) per Gb sequenced. Biomes were compared using Benjamini-Hochberg FDR-corrected pairwise ranked comparisons following Kruskall-Wallis, and common letters to the right in each boxplot denotes biomes with similar median gene prevalences. Numbers to the right denote the number of metagenomes included for each biome. Box boundaries represent first and third quartiles, with midline denoting the median and whiskers the 1.5 IQR. Mean composition of clade I (**c**) and clade II *nosZ* reads (**d**), organized by biome and colored by phylum. Biomes represented by fewer than 20 metagenomes are excluded from the figure. The percentage of genomes having clade II *nosZ* and a complete denitrification pathway are indicated for each phylum in the legend and can be seen for both clades in Supplementary Fig. 2. Standard errors corresponding for each phylum are available in Supplementary Data 12. Host assoc. host-associated, Engineer. engineered. Source data are provided as a Source Data file.

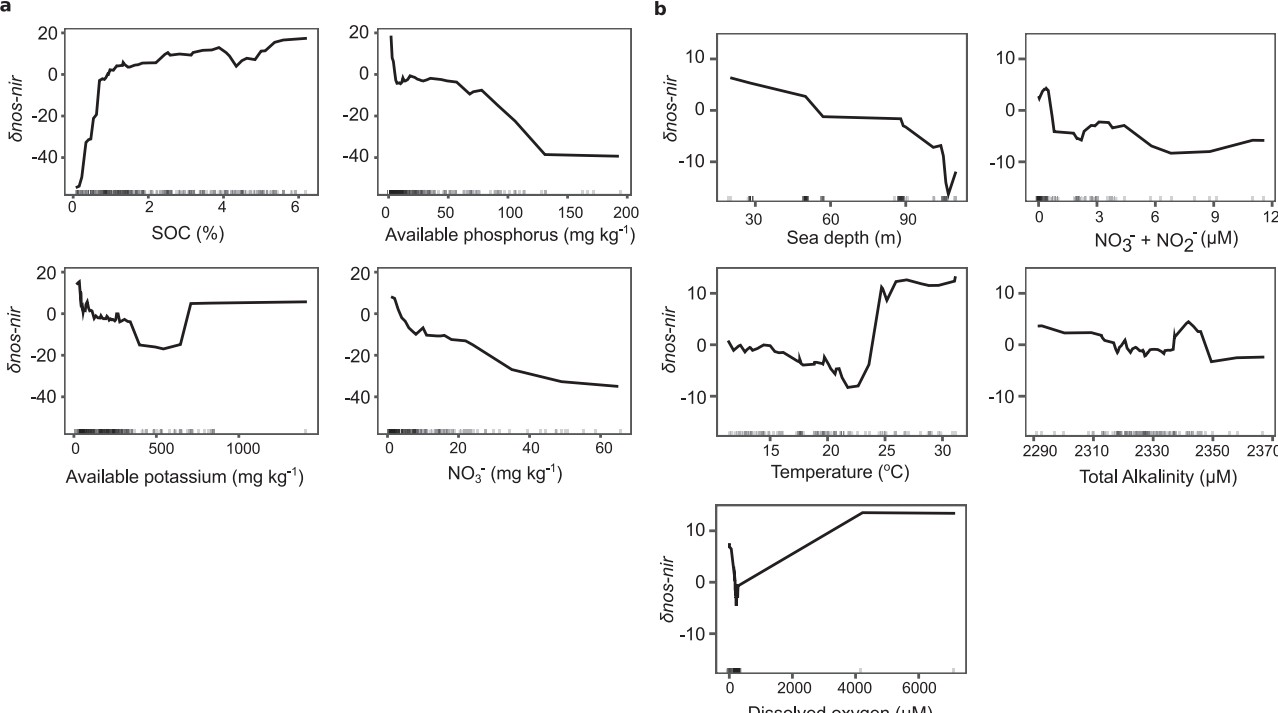

**Fig. 4 | Environmental predictors of balance between *nos* and *nir* counts using random forest models.** Abiotic predictors of the difference in *nos* and *nir* gene counts, calculated as (*nosZ* I + *nosZ* II) − (*nirK* + *nirS*) and normalized per Gb sequenced in soil (**a**) and marine metagenomes (**b**). Accumulated local effects plots show the differences in prediction of the δ*nos-nir* (y-axis) compared to the mean prediction along the range of each predictor (x-axis), while accounting for potential correlations among predictor values. Values above zero indicate the model predicts higher than average dominance of *nosZ* over *nir* at a given value of the predictor variable. The analyses were performed on a subset of the soil (n = 298) and marine (n = 89) metagenomes for which relevant environmental metadata was available (see "Methods"). Root mean square error and variance explained were 37 and 27% in the soil (**a**) and 19 and 26% in marine (**b**) models, respectively. Vertical marks on the x axis in each panel denote data density. Five hundred trees were run, and mtry, nodesize, and sampsize parameters were 4, 8, and 238 for soil and 2, 3, and 71 for the marine models, respectively. Source data are provided as a Source Data file.

prevalent compared to *nir* as available phosphorus increased, indicating that high nutrient conditions in soils favor $NO_2^-$ reducers over $N_2O$ reducers. Within the marine dataset, predicted δ*nos-nir* was lower at the highest concentrations of the sum of $NO_3^-$ and $NO_2^-$ compared to the lowest concentrations, and showed a dramatic shift in prediction from a decreasing to increasing relationship with temperature at 22 °C (Fig. 4b).

Available $NO_3^-$, phosphorus, potassium, and zinc had a negative effect on predicted δ*nosZ*I-*nosZ*II (Fig. 5a), which suggest that partial $N_2O$ reducers increase with increasing nutrient levels. pH and clay content also had a negative effect on predicted δ*nosZ*I-*nosZ*II, though only pH had a negative effect over the entirety of measured values. Increases in elevation at low altitude were associated with increased predicted δ*nosZ*I-*nosZ*II, but additional increases in elevation were not associated with further increases. In the marine study, temperature and ammonium concentrations had opposing unimodal effects on predicted values over their observed ranges (Fig. 5b).

### Expression of denitrification termination and initiation in the environment

We subsequently quantified the presence of *nosZ* and *nir* transcripts in 413 metatranscriptomes from soils and aquatic environments, distributed among five biomes, to assess whether the biome-level differences and associations between environmental variables and δ*nosZ*I-*nosZ*II and δ*nos-nir* gene abundance were also apparent in gene expression. *nirK* transcripts were strongly dominated by archaeal nitrifier reads (median 50%, range 0–100%), which we excluded in our analysis (see "Methods"). Consistent with the metagenome analysis, *nir* was more prevalent than *nosZ* and clade II *nosZ* was more prevalent

than clade I *nosZ* (Fig. 6a). The soil studies either directly or indirectly manipulated carbon availability, and soils associated with higher carbon availability had higher δ*nos-nir* expression than their paired lower-carbon samples (SMD 0.75, p = 0.015) but did not have higher δ*nosZ*I-*nosZ*II (SMD 0.06, p = 0.73). Among aquatic studies, which were all observational, correlation coefficients were negative between δ*nos-nir* and the $NO_3^-$ + $NO_2^-$ (mean −0.27, CI: −0.53 to 0.00) and positive between δ*nos-nir* and chlorophyll concentrations (0.22, CI: 0.04, 0.41; Fig. 6b). δ*nos-nir* was not correlated with dissolved oxygen or phosphate concentrations, or with bacterial production, another proxy for resource availability. δ*nosZ*I-*nosZ*II was positively correlated with dissolved oxygen content, which would occur if the dominant organisms encoding clade I NosZ lowered oxygen concentrations around their $N_2O$ reductase better than those encoding clade II[43].

## Discussion

By combining a comparative analysis of genomes with broad metagenomic and metatranscriptomic surveys, we provide fresh insight into the prevalence of the complete and partial denitrification trait among global microbial communities. In contrast to a previous comparative genomics study based on a small set of isolated microbes[4] but consistent with more recent studies of MAGs from different environments[5,9,44–47], we found that the genetic potential for complete denitrification is less common than that for partial denitrification, among both genome-sequenced microorganisms and environmental communities from all major biomes. Our results show denitrification is primarily a community trait based on division of labor, thereby relying cross feeding between microorganisms producing and consuming $NO_2^-$, NO and $N_2O$ to complete the pathway. Furthermore, other genes

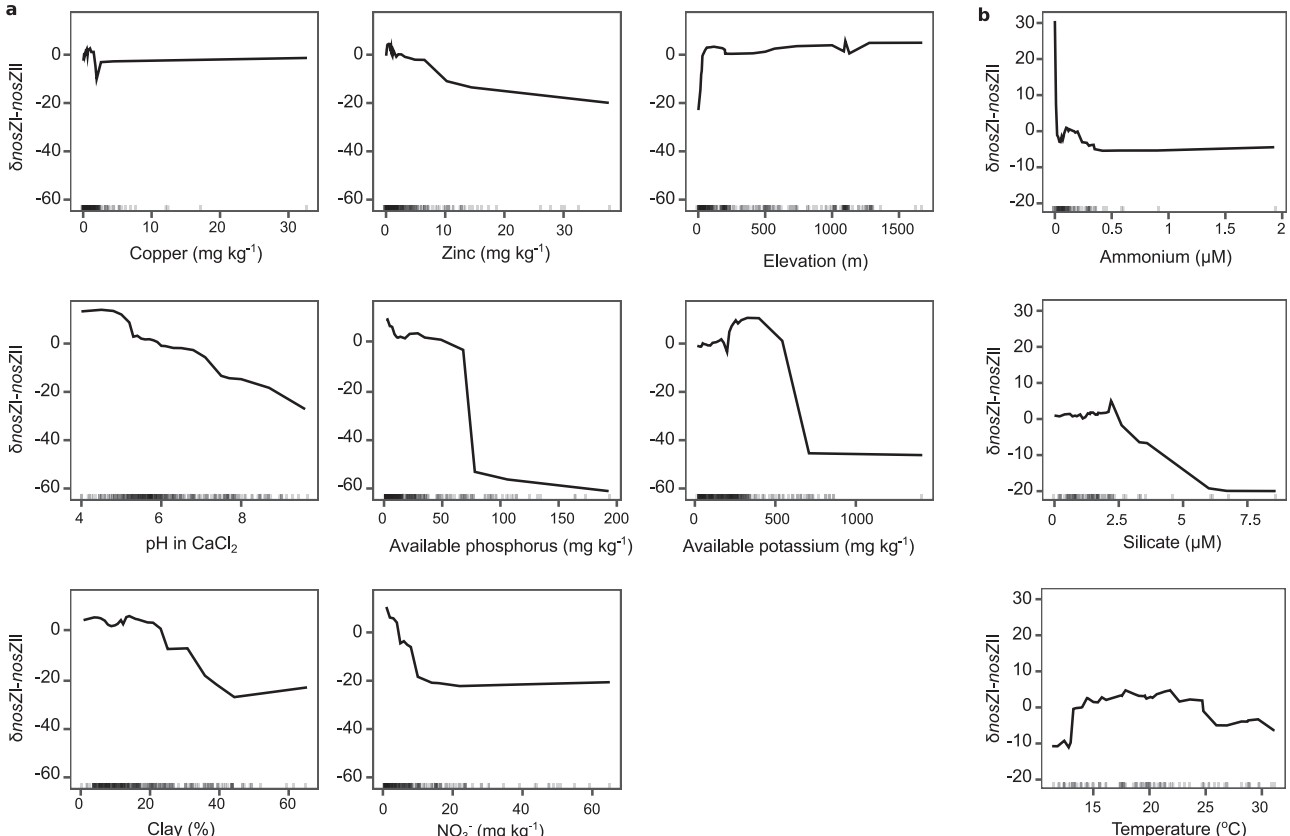

**Fig. 5 | Environmental predictors of balance between *nosZ* clade I and clade II using random forest models.** Abiotic predictors of the difference in *nosZ* clade I and II gene counts normalized per Gb sequenced in soil (**a**) and marine metagenomes (**b**). Accumulated local effects plots show the differences in prediction of the δ*nosZ*I-*nosZ*II (y-axis) compared to the mean prediction along the range of each predictor (x-axis), while accounting for potential correlations among predictor values. Values above zero indicate the model predicts higher than average dominance of *nosZ* clade I over *nosZ* clade II at a given value of the predictor variable.

The analyses were performed on a subset of the soil (n = 298) and marine (n = 89) metagenomes for which relevant environmental metadata was available (see "Methods"). Root mean squared error and variance explained were 41 and 58% in the soil (**a**) and 14 and 30% in marine (**b**) models, respectively. Vertical marks on the x axis in each panel denote data density. Five hundred trees were run, and mtry, nodesize, and sampsize parameters were 7, 8, and 238 for soil and 2, 10, and 62 for the marine models, respectively. Source data are provided as a Source Data file.

and genomic traits implicate complete denitrifiers as metabolically flexible generalists compared to any partial denitrifier type.

Various explanations exist for the dominance of partial denitrifiers. We show that partial denitrification enables greater overall niche partitioning, which was supported by narrower substrate ranges, lower capacity for oxidizing and reducing various inorganic electron donors and acceptors, and a lower genomic density of transcription factors and transporters among partial denitrifiers compared to complete denitrifiers. In the environment, higher *nirK* phylogenetic diversity, an indication of greater functional diversity among nitrite reducers[25], was associated with an increasing balance between capacity to initiate and terminate denitrification at the community level. This would further support niche partitioning if the pathways were split among partial denitrifiers, which is likely considering that that we mainly observed negative δ*nos*-*nir* values in the environment and the fact that MAGs are predominantly (84%) partial denitrifiers. Partial denitrifiers may also dominate communities due to the presence of a rate-efficiency tradeoff, i.e., that energy flux is slower through long pathways but potentially enables more complete usage of the terminal electron acceptor when it is limiting[18]. However, the dominance of partial denitrifiers among $N_2O$ reducers in the environment was not coherently explained by a rate-efficiency tradeoff. If this were the case, we would expect complete denitrification to decline as $NO_3^-$ increases or as $C:NO_3^-$ and $C:NO_2^-$ ratios decrease, assuming denitrifiers are predominantly heterotrophs in the environment. This agrees with

predictions based on modeling and observations in oxygen minimum zones in the oceans, proposing that the prevalence of a complete pathway increases as the limiting substrate shifts from C to N[20]. Accordingly, our random forest modeling inferred that the $N_2O$ reducing community in higher $NO_3^-$ soils became more dominated by partial denitrifiers, i.e., those carrying *nosZ* clade II, and fertilized croplands had more negative δ*nosZ*I-*nosZ*II than other soils in our cross-biome study. However, a similar reduction in δ*nosZ*I-*nosZ*II was not associated with increasing $NO_3^- + NO_2^-$ in aquatic metagenomes and metatranscriptomes or in the analysis mapping reads to marine MAGs. In addition, there was no relationship between increasing carbon content in soil or higher chlorophyll concentrations in aquatic samples and higher prevalence of complete denitrifiers in either metagenomes or metatranscriptomes. This is consistent with a recent qualitative analysis of ~1600 MAGs[44], which concluded that a rate-efficiency tradeoff cannot explain observed partial denitrifier dominance. Based on our findings of broader substrate ranges and greater genomic allocation to substrate uptake and transcriptional regulation, we instead propose that complete denitrifiers are adapted to flexibly take advantage of resources varying in time and space, while partial denitrifiers are restricted to slower and therefore less variable growth rates. This may explain the over-representation of complete denitrifiers among model denitrifiers and isolates that grow readily under standard resource-rich lab conditions. Under carbon-rich conditions supporting rapid growth in the environment, complete denitrifiers

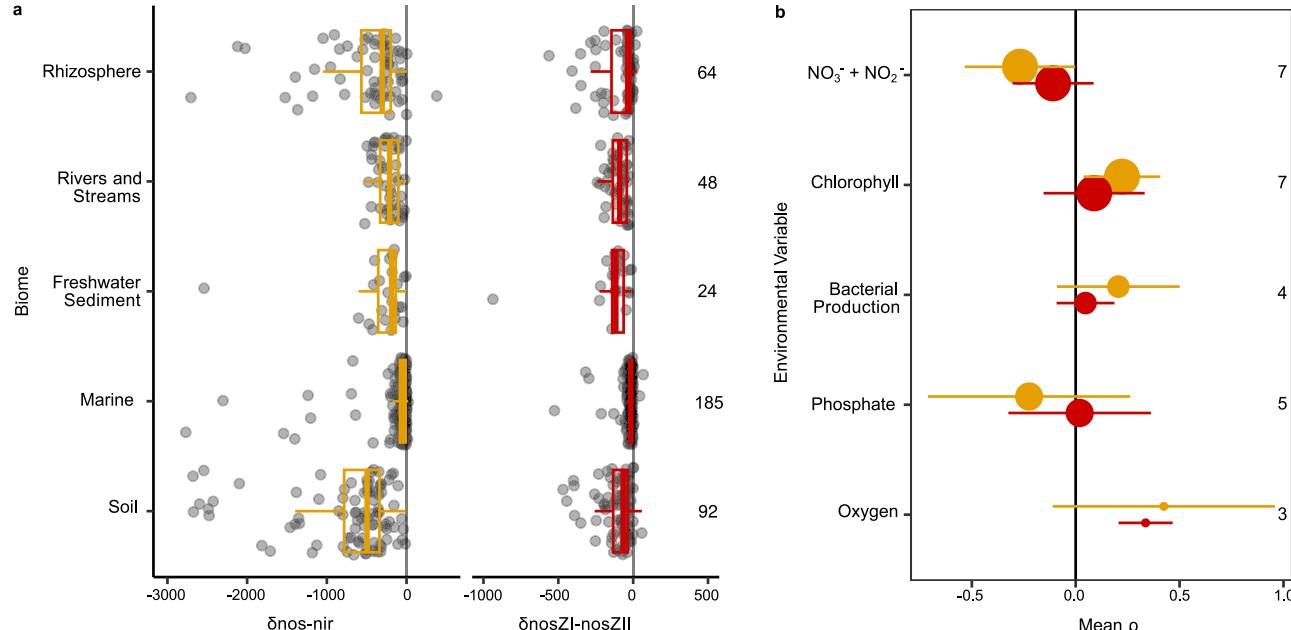

**Fig. 6 | Balance in prevalence of denitrification gene transcripts shows dominance of initiators at the community level in terrestrial and aquatic biomes. a** δ*nos-nir* and δ*nos*ZI-*nos*ZII normalized per Gbp mRNA sequenced across different biomes. Numbers to the right denote number of metatranscriptomes included in each category. **b** Mean and 95 % confidence intervals of spearman correlation coefficients between δ*nos-nir* or δ*nos*ZI-*nos*ZII and environmental factors in aquatic metatranscriptomes. Coloring follows gold for δ*nos-nir* and red for δ*nos*ZI-*nos*ZII as in (**a**). Circle size and numbers to the right denote the number of studies the effect size is calculated from. Soil samples with root exudate addition are categorized under rhizosphere in (**a**). Box boundaries represent first and third quartiles, with midline denoting median and whiskers the 1.5 IQR. Source data are provided as a Source Data file.

may benefit from using available $NO_3^-/NO_2^-$ more efficiently[20], while under rapidly fluctuating conditions many have the flexibility to use whichever electron donor and terminal electron acceptors are most readily available, including oxygen via high affinity terminal oxidases. This indicates that partial denitrifiers dominate because they have cheaper metabolisms to run, that do not require such complex regulatory mechanisms. This is consistent with the overall dominance of slow-growing organisms in natural environments[31,48,49].

The community level genetic balance between initiation and termination of denitrification indicates a higher capacity for $N_2O$ production than reduction in nearly all biomes. The variations between biomes and among the specific ecosystems within biome categories were readily explained by environmental variables. We found that high $NO_3^-$ levels were associated with lower δ*nos-nir* in both soils and aquatic systems, indicating greater imbalance between the two genes and a decrease in the relative importance of $N_2O$ reducers. In soils, the same pattern was observed for phosphorous. High concentrations of $NO_3^-$ may also allow denitrifiers to outcompete ammonifiers, which both theoretical[50,51] and empirical evidence indicate prevail under high C:$NO_3^-$ [52,53], including in the soil metagenomes used in this study[42]. Overall, this shows that nutrient rich environments promote denitrifier communities dominated by initiators and similar patterns were observed at the transcriptional level in aquatic environments (no data for soils). The observed imbalance in the denitrifier community could explain why nutrient loaded environments like agricultural soils are major sources of $N_2O$. However, complete denitrifiers have been observed to express only enzymes involved in denitrification initiation under high $NO_3^-$ and/or $NO_2^-$ [54,55], which also lead to accumulation of $N_2O$. Furthermore, high fertilization rates commonly favor nitrifier-denitrification[56], and hydroxylamine from nitrification and high $NO_2^-$ concentrations can both induce $N_2O$ production from chemodenitrification[56,57]. Finally, $N_2O$ may also be produced during the detoxification of NO from host immune responses[58,59]. Therefore, there are a wealth of pathways that are distinct from denitrification and

can reduce $NO_2^-$ and/or produce $N_2O$, and it is plausible that inter-pathway competition for $NO_2^-$ favors the presence of initiators ready to consume $NO_2^-$ in denitrification rather than letting it enter other pathways. Terminators could conceivably take advantage of the accumulated $N_2O$ from all these possible sources, in particular obligate terminators (i.e., predominantly *nosZ* clade II) which use $N_2O$ as terminal electron acceptor and thereby serve as important $N_2O$ sinks[16,60]. This reasoning agrees with the observed low δ*nos*ZI-*nos*ZII in metagenomes from agricultural soils and wastewater treatment plants. We conclude that high levels of $NO_3^-$ allow both initiation and termination specialists to prevail, but with a dominance of initiators which altogether increase the risk of $N_2O$ emissions.

Fixed carbon availability was associated with reduced dominance of *nir* over *nosZ* in aquatic transcriptomes and soil metagenomes, in particular in soils with less than 1% SOC. The soil metatranscriptomes also indicated increased carbon availability could help NosZ overcome a low competitiveness for electrons[61-63]. However, the chemical composition of the available carbon is expected to affect the balance between initiation and termination of denitrification and complete vs. partial denitrifiers. Organic acids have short catabolic pathways and fewer opportunities for enzyme bottlenecks to slow the flow of electrons into the electron transport chain compared to sugars and should favor complete denitrifiers able to use an array of terminal electron acceptors, in agreement with our comparative genome analysis. At the community level however, organisms metabolize and co-metabolize[64] substrates from among tens of thousands of molecularly-distinct potential electron donors of varying bioavailability[65,66], leading to conflicting effects of sugar and organic acids addition on $N_2O$ to $N_2$ production from soils[67-69]. High dissolved oxygen concentrations were also positively associated with dominance of clade I over clade II *nosZ* I expression but not gene abundance in aquatic samples. There is no clear consensus on clade I being less sensitive to high oxygen than clade II NosZ[70,71], and oxygen niche of $N_2O$ reducers may be a function of organism oxygen consumption[43] or even indirectly their

competition for $NO_2^-$ with nitrite oxidizing bacteria[20]. Thus, while the mechanism for oxygen preference cannot be inferred from our data, our results nonetheless indicate that terminators and complete denitrifiers make distinct contributions to $N_2O$ reduction as a function of environmental conditions.

Our results indicate that the prevalence of denitrification functional types within and between environments should be considered in light of not only the potential organism level advantages to having a particular denitrification genotype under given environmental conditions, but also through biotic interactions. Denitrification does not occur in isolation, and its steps may occur in response to its intermediates being produced by other pathways. Further, spatiotemporal variation in electron donor and acceptor ratios can enable coexistence of organisms completing different steps of denitrification. Future work addressing phenotypic plasticity of denitrifiers under realistic environmental conditions is necessary to resolve this uncertainty and disentangle biotic and abiotic drivers of denitrifier community assembly. This will be particularly enhanced by improved generation of high-quality assemblies of representative environmental genomes, and by the ability to sample microbial communities on the scale at which they interact with one another and the environment.

## Methods

### Denitrification enzyme databases

Alignments and phylogenies for NirK and NirS were obtained from Pold et al.[25]. The Nor reference database was derived from Murali et al.[24] and consisted of 67 Nor sequences (cNOR, qNOR, bNOR, eNOR, gNOR and sNOR, and nNOR) and 865 other heme copper oxidases. A hidden Markov model for NosZ was generated downloading all bacteria and archaea genomes from NCBI (7 October 2021) and using hmmsearch v.3.2.1[72] to identify NosZ using an HMM built from the alignment from in Graf et al.[73]. DNA sequences were translated to amino acids before being dereplicated at 100% identity with CD-HIT (v 4.6[74]). After initial maximum likelihood (ML) trees reconstruction with FastTree v. 2.1.11[75], removal of sequences derived from genomes either identified as contaminated in the GenBank metadata or as <80% complete or >5% contaminated by BUSCO v. 5.3.1[76], we manually checked the alignment for conserved residues, aligned poorly-conserved regions of the sequence, trimmed the alignment to its conserved core in Arb v.7.0[77], and built a phylogeny using IQ-TREE v. 2.1.3 with the best model identified by modelfinder (LG + R10)[78]. Nitrate reductases, catalyzing the first step in the denitrification pathway, were excluded from the analyses because they are found in a wide variety of microorganisms and their activity often occurs in isolation from the characteristic gas-producing denitrification process itself[23,79].

### Genome annotation

We used the representative set of genomes and associated genome quality metadata from GTDB v214.1 for our analysis (Supplementary Data 2[80]). Genomes less than 80% complete, more than 5% contaminated, or which failed taxonomic consistency of contigs based on GUNC v.1.0.6[81] were excluded from our analysis, resulting in 61,293 genomes in our final dataset. We included both isolate (49% of assemblies) and culture-independent genome assemblies (metagenome assembled and single cell genomes; 51%) in our analysis to balance genome quality with inclusion of genetic diversity representative of environmental samples. We identified genes for NirK, NirS, Nor, and NosZ in each genome using an HMMsearch[72] against the corresponding amino acid databases as described above. After aligning potential proteins to the reference, we imported them into ARB v.7.0[77] and used a combination of phylogeny building with FastTreeMP v.2.1.11[75] and manual curation to exclude reads lacking conserved ligand binding domains. Nor and NosZ were also classified according to protein class (i.e., bNOR, cNOR, eNOR, gNOR, nNOR, qNOR and sNOR for Nor and halophiles, clade I and clade II for NosZ). We then classified each

genome into non-denitrifiers (lacking *nir* and *nosZ*), complete denitrifiers (*nir*, *nor*, and *nosZ*), and partial denitrifiers (lacking *nir* or *nosZ*), with partial denitrifiers further divided into initiators (*nir* with or without *nor*), terminators (*nosZ* with or without *nor*) and initiator-terminators (*nir* and *nosZ*; Fig. 1b). Anammox genomes carrying *nir* (n = 47) were classified as non-denitrifiers because their nitrite reductase is not thought to be involved in denitrification[82]. Similarly, genomes encoding just *nor* were denoted as non-denitrifiers (n = 3064) as a conservative measure since they are likely involved in detoxification rather than respiration. Since we only used bacterial genomes, our analysis includes ammonia oxidizing bacteria potentially capable of nitrifier-denitrification, but not ammonia oxidizing archaea for which a respiratory function for nitrite reductase has not yet been established[83].

A range of tools were used to identify and validate genes associated with potential electron donors and acceptors used by microorganisms. Genes for proteins involved in arsenite oxidation (AioBA; validated with alignment from Quemeneur et al.[84]), anammox (Hzs, Hdh) and iron redox were identified in genomes using MagicLamp v.1.0 with final annotations curated to match minimum subunit composition proposed for each enzyme[85]. We used AmoA as a marker for ammonia oxidation in bacteria using an HMM built from the alignment of AmoA, HmoA and PmoA from Diamond et al.[5]. Respiratory reductive dehalogenases were identified using the RDaseDB alignment (v.2020[86]) combined with a literature search to exclude catabolic enzymes; we note that the TAT motif proposed to be indicative of respiratory enzymes was absent from the majority of sequences and could not be used. Sulfur redox genes were identified using HMSS2 and categorized into oxidation or reduction-associated enzymes following Tanabe and Dahl[87]. Sulfide: quinone reductase (SQR) enzymes were excluded from our analysis, both because their physiological role is variable, and because we found the HMMs were unspecific. Hydrogenases were identified using MagicLamp v.1.0, refined using the residues in Greening et al.[33], and verified as to their clade and probable physiological function using the HydDB web server[88]. Clades 1a−d, 1 g, h and j and clade 2 were considered involved in hydrogen oxidation, and clades 4a, b, c, and h were considered involved in reduction. ArrA and ArxA involved in arsenate reduction and arsenite oxidation, respectively, were identified using an HMM built from sequences from Wells et al.[89], and validated using the residues from Ospino et al.[90]. Genomic potential for aerobic respiration was identified by screening the HMM-based HCO family search used to annotate Nor for HCO classes A, B and C, and completing a second HMM-based search using the cytochrome *bd* oxidase family reference alignment from Murali et al.[91]. The proteins were categorized into high (HCO classes B and C and cytochrome *bd* oxidase) and low affinity oxidases (HCO class A)[92]. All redox gene annotations are reported in Supplementary Data 3.

Sugar-acid preference of organisms was inferred using the sum of KEGG orthologues identified by enrichM (v0.6.3, default settings) that were annotated to sugar vs. acid metabolism[32], noting that the majority of the genomes fall outside the originally-calibrated range of GC contents (Supplementary Data 4). GapMind carbon[93] (March 22nd, 2021 database version) was run to infer ability to grow on 62 compounds. We used custom pathway completeness cutoff scores for each pathway based on annotations for organisms for which growth had been reported on that substrate in BacDive (Supplementary Data 5). Our positive controls dataset did not have data for 6 substrates, leaving 56 for analysis. Transporters were annotated using DIAMOND (2.1.6.160[94]) against the TCDB database (download date: 16th May 2023)[95], with hits showing at least 40% identity to the reference and 70% coverage of both reference and query sequence, with no more than 10% difference in alignment length kept for analysis[96]. Transcription factors were annotated using DeepTFactor[97] (v. 2020-11-09) with default settings (Supplementary Data 6). We used GenomeSPOT v.1.0.1 to infer temperature, pH, and salinity ranges[98], and used the

difference between the maximum and minimum values for each genome in our analysis (Supplementary Data 7).

Unless otherwise noted, all subsequent analyses were completed in R v.4.1[99]. The maximum growth rate of each taxon was estimated using gRodon2[31], which used codon usage bias of ribosomal proteins annotated in the genomes using hmmearch against the ribosomal proteins extracted from Uniprot PFAM v. 35.0 with profile-specific cutoffs (Supplementary Data 8[100]). Differences in growth rates between denitrifier types were assessed using Kruskall-Wallis tests (stats::kruskal.test) followed by Dunn tests with Benjamini-Hochberg correction for multiple testing (FSA::dunnTest), where all maximum growth rates less than 0.2 doublings h$^{-1}$ were given the same rank. This cutoff was needed because codon use bias saturates with growth rates slower than this[31].

To determine whether complete and partial denitrifiers differ in transcription factor, transporter, carbohydrate-active enzyme, or catabolized substrate count or sugar acid preference, we fit linear models, logging or taking the exponent of variables where needed. However, we analyzed GapMind data with a generalized linear mixed model with a Tweedie distribution in glmmTMB[101] (v.1.1.10) due to the poor fit of OLS and high proportion (10-37%) of assemblies where no substrates within a chemical class were predicted to be used. We subsequently ascertained that the model residuals were not phylogenetically clustered using phytools::phylosig (v.2.3.0[102]) and an organism tree derived from the GTDB reference tree[80]. This tree was rerooted in Fusobacteriota[103] and forced to be ultrametric using the castor::date_tree_red function[104] (v.1.8.2). Blomberg's K[105] of model residuals was less than one in all instances (P < 0.05), so we did not include phylogenetic relatedness as a random effect in our analysis. We evaluated differences between complete and partial denitrifier genome content using post-hoc multiple comparisons with the sandwich::vcovHC function (v.3.1.1[106]) to account for heteroskedasticity. Due to the large number of zeroes, we assessed the hypothesis that partial denitrifiers encode the potential to oxidize and reduce a smaller number of inorganic compounds than complete denitrifiers by comparing the cumulative density distributions using a one-sided Kolmogorov-Smirnov test using stats::ks.test.

**Metagenome and metadata collection**
To assess the dominance of complete vs. partial denitrifier communities across biomes, we selected metagenomes with at least 100,000 reads ≥150 nt, primarily from larger sampling campaigns where uniform metadata were available (Supplementary Data 9). The majority of soil metagenomes came from the Australian Microbiome Project/BASE[107], the National Ecological Observatory Network[108], Topsoils Microbiome Project[109], Long-Term Soil Productivity experiment[110], and the Stordalen Mire[111]. The majority of aquatic metagenomes came from the Australian Microbiome Project[112], Linnaeus Microbial Observatory[113], the Amazon Continuum Project[114] and BATS, GEOTRACE and HOTS[115]. Engineered metagenomes included those from drinking water[116] sewage[41], and wastewater treatment plants[117]. The host-associated metagenomes dataset consists primarily of plant-associated sequences, such as from leafy greens[118], beans[119], citrus rhizosphere[120], switchgrass[121] and *Arabidopsis*[118], though we also included a handful of sequences from invertebrates such as sponges and gutless marine worms. For unpublished metagenomes where contact details were provided, we emailed authors to request permission to use them.

We completed a similar search for metatranscriptomes, focusing on soil (including rhizosphere) and marine studies where carbon and/or nutrient availability had either been manipulated or logic or metadata indicated they differed between samples (Supplemental Data S10). These metatranscriptomes were identified based on searching the American Society for Microbiology journals website for metatranscriptomes and google databases for the terms "soil" or "marine" and "carbon" or "nutrient" and "metatranscript*".

Where multiple sequencing runs or files existed for the same sample, we combined the fastq files prior to searching. We used only the forward read in the case of paired end reads, and only the first 150 nt of any reads longer than that. In total, our dataset comprised 3991 metagenomes, with 1489 from aquatic habitats, 1642 from soil, 658 host-associated, and 202 from engineered habitats. Our final set of metatranscriptomes overlaps minimally with our metagenome dataset and contains 413 metatranscriptomes (64 rhizosphere or soil with root exudates added, 92 other soil, 185 primarily coastal and estuarine marine, 24 lake sediment, and 48 river metatranscriptomes).

Biomes were assigned to metagenomes based on The Nature Conservancy Terrestrial Ecoregions for terrestrial samples[122] and latitude for marine samples (polar: latitude > 60°; westerlies: latitude 30–60°; trades: latitude 0–30°). Terrestrial biome assignment required use of the packages sp v.1.4-5[123], rgdal v.1.523[124], and rgeos v0.5-5[125]. Soils under cultivation were excluded from the biome-based approach and instead categorized as croplands.

**Dominance of complete vs. partial denitrifier communities based on proxy gene searches across biomes**
We selected mRNA reads from metranscriptomes using SortMeRNA v.4.3.7 with the smr_v4.3_default_db database and settings "--other --fastx --num-alignments 1 --no-best". We used GraftM v.0.13.1 to identify denitrification genes (*nirK*, *nirS* and *nosZ*) in the metagenomes and metatranscriptomes, as previously described[25]. Briefly, GraftM uses a two-step process in which a HMM search identifies candidate reads, followed by phylogenetic placement on a reference tree using pplacer (v.1.1.19[126];). We used the accumulate function in gappa v.0.8.0[127] to find the position on the tree where at least 95% of the mass for each read descended from, and excluded all reads with any mass placed in the outgroup. We further excluded sequences placed in the non-denitrifying anammox NirS clade 1 h (median 0% of *nirS* reads in metagenomes and 0% in metatranscriptomes), archaeal nitrifier NirK clades 2 and 4 (1.4% of *nirK* reads in metagenomes and 50% in metatranscriptomes), as well as the eukaryotic NirK clades 1b and 1e (0% and 0%)[25]. Our analysis also excluded the reads placed in the recently reported clade III *nosZ*[128], which forms the outgroup in our reference tree. This metagenome and metatranscriptome search depended on: hmmer (v.3.2.1[72];), OrfM (v.0.7.1[129];), bbtools (v.38.90; https://sourceforge.net/projects/bbmap/), lbzip2 (v.2.5; https:// lbzip2.org/), and fxtract (v.2.3; https://github.com/ctSkennerton/fxtract).

We validated these search and placement and post-processing parameters by generating 150 nt fragments of full-length sequences that were picked up by the NirK, NirS and NosZ HMMs, which in addition to the target proteins also included homologous multicopper oxidase proteins from Cyanobacteriota and Thermoprotetoa, NirN and NirF, and various members of clade III NosZ, respectively. Sensitivity is the fraction of total ingroup fragments searched which were placed in the ingroup, and specificity is one minus the fraction of outgroup fragments incorrectly placed in the ingroup (or, in the case of *nosZ*, in a clade other than the target clade). Sensitivity was 76% for *nirK*, 93% for *nirS*, 91% for clade I *nosZ* and 94% for clade II *nosZ*. Specificity was 97% for *nirK*, 100% for *nirS*, 100% for clade I *nosZ* and 100% for clade II *nosZ*. These specificity scores do not preclude the possibility of reads from other related proteins not included in these outgroups from being placed within the phylogeny.

We compared gene prevalence across biomes using the absolute difference in counts of clade I and II *nosZ* or *nir* and *nosZ* genes standardized by the number of Gbp sequenced (or Gbp mRNA sequenced for metatranscriptomes) to account for differences in sequencing depth. For *nir* vs. *nosZ* abundance we used Eq. 1 such that positive values denote *nosZ* is more abundant than *nir*:

$$\delta_{nos} - nir = 10^9 x \, ((nosZI + nosZII) - (nirS + nirK))/(150 \text{ x reads}) \quad (1)$$

For *nosZ*, we used Eq. 2 such that positive values denote clade I is more abundant than clade II:

$$\delta_{nos}ZI - n_{os}ZII = 10^9 \text{x} (nosZI - nosZII)/(150 \text{ x reads}) \quad (2)$$

This method of calculating gene prevalence enables the abundance of both genes to be considered concurrently. This is particularly relevant for the 14% of metagenomes and 26% of metatranscriptomes where *nosZ*I was not detected but *nosZ*II was found in 1–13,041 (1–10,926 in metatranscriptomes) copies, and the 0.8% of metagenomes where *nosZ* was not found but 3-28 copies of *nir* were identified.

## Environmental correlates with biome gene counts and denitrifier MAGs

We used the TARA Oceans[35] MAGs present in the OceanDNA MAG database to test the hypothesis that complete denitrifiers are more abundant under conditions with high electron donor to acceptor ratios. These MAGs were chosen because the database is sufficiently large that multiple MAGs from each denitrifier type category were present, and where multiple conspecific MAGs were present, those with inconsistent denitrification gene repertoires could be excluded. After quality-controlling the raw paired-end reads with cutadapt (-m 100 -q 20 –max-n 0 –trim-n), we mapped them to all MAGs using CoverM (v. 0.6.1[34]; -m relative abundance –min-read-aligned-percent 0.75 –min-read-percent-identity 0.95 –min-covered-fraction 0) then determined the total relative abundance of each MAG-OTU in each sample (95% ANI). Only the reads for the 3 μm filter fraction were used in this analysis to allow for bigger cells or cell aggregates than using the 0.22 μm fraction. We subsequently determined the standardized relative abundance by dividing the observed relative abundance within each sample to the maximum relative abundance observed across all samples. We then fit a linear model predicting the standardized relative abundance of each MAG OTU based on the combination of $NO_3^-$ and chlorophyll concentrations (i.e., $NO_3^-$ after accounting for chlorophyll, chlorophyll after accounting for $NO_3^-$, and the logarithm of chlorophyll: $NO_3^-$). The mean slope and standard error for each MAG OTU were extracted and used to calculate the weighted mean response and 95% confidence intervals for each denitrifier type. Differences in mean response to $NO_3^-$, chlorophyll and their ratios between denitrifier types were established based on non-overlapping 95% confidence intervals.

We evaluated the relationship between differences in normalized gene counts and resource availability using random forests. This analysis was restricted to metagenomes from the BASE and Australian marine projects[107,112] as examples of studies with a large number of metagenomes with uniformly-collected metadata for variables associated with denitrification, including $NO_3^-/NO_2^-$ levels, pH, carbon, and micronutrients. A pre-selection of variables for inclusion in the model was completed to exclude collinear variables (Spearman's $\rho > 0.7$ or variance inflation factor > 4); in the collinear groups, the variable hypothesized to be most strongly and directly explicable of gene ratio was retained (Supplementary Fig. 3; Supplementary Tables 2 and 3). We subsequently ran VSURF v. 1.1.0[130] 100 times to identify the best predictors for each ratio and biome, keeping only those variables retained in at least 95 of the 100 iterations. We then used the randomForest package v. 4.7.1–1 to model the relationship between gene ratios and the final set of retained variables. A grid search was used to find the combination of tuning parameters yielding lowest out-of-bag root-mean-square error. Accumulated local effects plots were generated to visualize results using iml v. 0.11[131].

Finally, we verified whether high carbon and/or low nutrient availability favored the transcription of *nosZ* over *nir* or clade I *nosZ* over clade II *nosZ* by calculating $\delta nos- nir$ and $\delta nosZI-nosZII$ in the metatranscriptomes, respectively. We calculated Hedges G in a meta-analysis of effect sizes for soil studies (meta::metacont v 8.2.1), taking the lower carbon soil as the reference and carbon-enriched soil as the treatment (e.g., bulk soil or unamended soil compared to rhizosphere, glucose, glycine, or root exudate amended soils). Aquatic studies were observational, so we fit a Spearman correlation coefficient between $\delta nos-nir$ or $\delta nosZI-nosZII$ and environmental variables ($NO_3^- + NO_2^-$, phosphate, bacterial production and chlorophyll A concentrations as proxies of resource availability, and oxygen). Where communities were captured and analyzed on two different filter sizes from the same sample, we determined the correlation separately for the two size fractions but counted them as a single study in our overall effect size calculations. We used psychmeta::ma_r (v2.7.0) to calculate overall correlations in the aquatic data.

## Reporting summary
Further information on research design is available in the Nature Portfolio Reporting Summary linked to this article.

## Data availability
The annotation data generated in this study are provided in the Supplementary Information. All genomes, metagenomes, and metatranscriptomes used are publicly available in NCBI or other sources under the identifiers denoted in Supplementary Data 2, 9, and 10. Hidden Markov Models, and reference databases used for denitrification gene searches are available in FigShare (https://doi.org/10.6084/m9.figshare.23913078 for NirK and NirS and https://doi.org/10.6084/m9.figshare.30122335 for NosZ). Source data are provided with this paper.

## Code availability
Scripts used to generate figures and annotate genomes are available in FigShare under (https://doi.org/10.6084/m9.figshare.30122335).

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

## Acknowledgements

This work was supported by the Swedish University of Agricultural Sciences Senior Career Grant 2019-2025 to S.H. and the Swedish Research Council Grant Agreements No. 2016-03551 to S.H. and 2023-03627 to G.P. Computing resources were provided by the Department of Forest Mycology and Plant Pathology, Swedish National Infrastructure for Computing project SNIC 2022/22-1119 and NAISS 2023/22-1278, funded by the Swedish Research Council through Grant Agreement No. 2022-06725, and the Swedish National Infrastructure for Computing (SNIC) at the PDC Center for High Performance Computing, KTH Royal Institute of Technology, partially funded by the Swedish Research Council through grant agreement no. 2016-07213 (NAISS 2024/22-133).

## Author contributions

Conceptualization: S.H. Data curation: G.P., A.S. and C.M.J. Formal analysis: G.P. Funding acquisition: S.H. and G.P. Investigation: G.P. Methodology: G.P., A.S., C.M.J., and S.H. Project administration: G.P. and S.H. Resources: S.H. Software: C.M.J., G.P., and A.S. Visualization: G.P., A.S. Writing—original draft preparation: G.P., A.S., and S.H. Writing—review and editing: G.P., A.S., C.M.J., and S.H.

## Funding

## Competing interests

The authors declare no competing interests.
