## [Transparent Peer Review file · Nature Communications]

Denitrification is a community trait with partial pathways dominating across microbial genomes and biomes

Corresponding Author: Professor Sara Hallin

Version 0:

Reviewer comments:

Reviewer #1

(Remarks to the Author)

Denitrification is a multi-step respiratory pathway wherein reduced products of nitrate or nitrite serve as electron acceptors in subsequent steps. Each step can operate and contribute to energy transformation independently, thus creating the potential for partial denitrification. This manuscript describes a large scale (meta)genomic survey of denitrifying prokaryotes across various biomes with attention to: (1) representation of partial vs full denitrifier groups and (2) correlation of these groups with predicted growth rate, biome, and available resources measured in those environments. The study is impressive in its scope and its attempt to correlate group representation with other parameters. As is inevitable that interpretation of datasets like this will be largely speculative but I think it will still be a valuable resource for the community.

As someone who understands the biology but is less familiar with working with such large data sets, I can offer that (1) I found the data to be insightful and I foresee that it would be a useful reference for the field but also that (2) the data could be made more approachable for readers that are less familiar with the approaches used.

Main comments:

1. What do the authors think of the absence of representatives with only *nor* genes? Is this surprising given the possible need to detoxify *NO* aside from using *NO* for respiration?
2. Section starting at L149 – this was one of the more confusing parts of the manuscript for me because:
 - a. Perhaps due to the use of 'could,' it sounds like you could do something if you had the data (L149), but then the data wasn't available (L153), but then proxies were used. If I replace 'we could make a more direct assessment' with 'we attempted to make a more direct assessment' then I think the progression of statements sounds less round-about. There might be other examples like this (e.g., on L163 'could calculate' vs 'calculated').
 - b. I don't work with environmental data sets like this but using chlorophyll as a proxy seemed problematic to me. I would expect it to correlate with electron donors but perhaps correlate even more with *O2* levels. The problem I see is where dissolved *O2* is used as a proxy for *NO3-*, assuming a negative correlation. Is interdependence between these parameters not an issue? Would it be easy to show data to illustrate that these are independent proxies?
 - c. I wonder if Fig 3 should be deprioritized, or even omitted, given the above and considering that the correlations were weak compared to those with the environmental parameters in Figs 5 and 6.
3. I recommend giving another proofread with a focus on how some sentences might be rephrased to facilitate understanding. I give a few specific examples below but there are more sentences that need attention.
4. It might help to comment on what assumptions are being made when interpreting the data. For example, partial denitrifiers might be less versatile/flexible with respect to nitrogen oxides but we are blind to what other metabolic capabilities are available for each organism (e.g., other respiratory pathways, fermentation, phototrophy; L257).

Other comments:

5. The sequential order of complete vs various subgroups of partial denitrifiers should be made consistent between figures.

6. Similarly, try to describe panels in the order they appear (e.g., L235).
7. L184 vs L187 Maybe better to use only 'difference' instead of introducing 'delta.'
8. L227 – 'up to' vs 'above'
9. L231 – I don't see NO₂- concentrations shown
10. L232 – confusing sentences describing a 'dramatic shift in prediction.' Maybe describe the trend more explicitly and give the temperature data its own sentence.
11. L258 – how is stability of growth rate inferred?
12. Fig 5 and 6 – I didn't see an explanation for the shaded squares along the x-axes.

Reviewer #2

(Remarks to the Author)

This study presents a meta-analysis of curated genomic data from isolates and ecosystems to co-locate known genes (using HMMs) involved in the steps of denitrification to determine the prevalence of partial versus complete pathways and the ecosystem-based variables that drive these trends. While the analyses here are a good step forward towards demonstrating that partial denitrifiers tend to dominate over complete denitrifiers in ecosystems, a trend that has been alluded to in other studies, there are some flaws and biases in the modeling exercise that limit the overall impact of the conclusions.

The analysis relies completely on *in silico* tools to determine prevalent types of denitrifiers within their niches based on how fast they grow or how they access resources. Thus, the data represent a model of microbial behavior and thus include the biases of algorithms and sampling schemes (e.g. heavy reliance on cultivated strain genomes). For instance, the gRodon tool used to predict growth rate is based on a presumed direct correlation between codon bias and growth rates in complete genomes; hence, this tool is less predictive of growth rates for the MAGs and SCGs relative to the complete isolate genomes. As such, the conclusions of this work present averages calculated across diverse types of data and do not necessarily reflect ecosystem or phylotype diversity in terms of ecosystem denitrifying potential.

One nagging question is how much would growth rate depend on metabolic flexibility, for instance facultative oxygen respiration in an ecosystem context? If there is a correlation between growth rate and complete denitrification, there are likely other physiological factors at play that are not purely related to denitrification. The model presented here has some gaps that lead to misconceptions about how denitrification and partial versus complete pathways are distributed based on niche. The model should be refined to include other factors that are known about complete versus partial denitrifiers, namely their ability to use oxygen, organic carbon, other key regulated pathways (e.g. switches between N₂O respiration and DNRA) in populations, and whether the genes come from lithotrophs (e.g. nirK in AOA) that would bias the data towards a partial denitrification phenotype. For instance, it is known that AOA can not denitrify but nearly all encode nirK and AOB encode nirK and norB genes. In other words, how much does metabolic diversity of the encoding microbial populations skew the generalized niche preference model presented in this study of partial and complete denitrifiers? Can this metabolic life-style diversity be accounted for (or discounted) somehow in the models presented here under "denitrification" when denitrifying genes are utilized by microorganisms in different metabolic contexts?

Could these models be tested and validated with gene expression datasets? It is not difficult to run the same HMMs on transcriptomes, and this could deconvolute the process differences (e.g. nitrification from denitrification)

line 73- For the introduction, it might be helpful to include a small figure showing the steps in the denitrification pathway to explain the definitions of different types (e.g. initiators, terminators, complete, and partial) as not all readers will be familiar with these pathways/processes.

line 164 - It doesn't seem surprising that denitrification would correlate with nitrate supply, since this is its primary substrate. This is essentially a confirmatory finding.

line 171 - if the genomes in the dataset are not representative of prevalent groups, then does that negate some of the averaged findings reported above? The lack of relevant, comprehensive, sampling and heavy reliance on cultivated strain genomes seems like a major problem for this study that aims to find a comprehensive model for denitrifier diversity.

line 195 - the hypothesis that nitrate assimilation has a higher rate than denitrification is testable as the authors could look at the ratio of nitrate assimilatory genes relative to denitrification genes.

line 209 - what is the rationale for why sewage communities for nosZ would be different than all of the other environments?

line 226 - if less than half of the variance in gene ratios was explained by the models, then shouldn't the models be better refined?

line 297 - why couldn't these additional drivers (e.g. organic matter) be included in the present study?

Version 1:

Reviewer comments:

Reviewer #1

(Remarks to the Author)

The authors have done a commendable job addressing reviewer comments. I have one lingering comment.

1. The new analysis of metabolic breadth was a welcome addition. However I was surprised by the choice of acceptors, which I would have expected to be phylogenetically constrained (though I could be wrong), while others were omitted:

- Why wasn't oxygen considered as a potential electron acceptor in Fig. 2e. I expect that aerobic respiration might be common and could correlate with other findings.
- The authors might also consider DMSO and TMAO, which might be important in marine environments. However, I understand that this is also getting away from the original intent of the manuscript.
- If you end up looking for the above acceptors, I don't suppose that fumarate reductase (for fumarate respiration) can be reliably distinguished from succinate dehydrogenase?

Reviewer #2

(Remarks to the Author)

The authors have meticulously responded to each of the reviewer comments and added additional data that clarifies and better validates their findings. In going through the revision, I do not see any major concerns that require comment.

Version 2:

Reviewer comments:

Reviewer #1

(Remarks to the Author)

Thank you for putting in additional work to address my lingering comment and my curiosity. I wish more authors approached reviewer comments like this group has done.

I agree with the revisions and the decisions the authors have made.

Reviewer #1 (Remarks to the Author):

Denitrification is a multi-step respiratory pathway wherein reduced products of nitrate or nitrite serve as electron acceptors in subsequent steps. Each step can operate and contribute to energy transformation independently, thus creating the potential for partial denitrification. This manuscript describes a large scale (meta)genomic survey of denitrifying prokaryotes across various biomes with attention to: (1) representation of partial vs full denitrifier groups and (2) correlation of these groups with predicted growth rate, biome, and available resources measured in those environments. The study is impressive in its scope and its attempt to correlate group representation with other parameters. As is inevitable that interpretation of datasets like this will be largely speculative but I think it will still be a valuable resource for the community.

As someone who understands the biology but is less familiar with working with such large data sets, I can offer that (1) I found the data to be insightful and I foresee that it would be a useful reference for the field but also that (2) the data could be made more approachable for readers that are less familiar with the approaches used.

Response: Thank you for the positive feedback. As recommended, we have tried to make the results more approachable.

Main comments:

1. What do the authors think of the absence of representatives with only *nor* genes? Is this surprising given the possible need to detoxify NO aside from using NO for respiration?

Response: We apologize for being unclear about the “*nor*-only” genomes. Representatives with only *nor* genes were actually not absent and represented 5 % of the genomes we analyzed. We categorized “*nor*-only” genomes as non-denitrifiers precisely because as the reviewer states, *Nor* can be involved in detoxification rather than respiration. The text has been revised to make this clear in the results (line 107; “Genomes carrying *nor* but not *nir* or *nosZ* accounted for 5% of the total number of genomes and were excluded in downstream analyses because *Nor* in *nor*-only genomes are likely involved in detoxification rather than respiration.”) and methods sections, as well as in the caption for figure 1b. The methods now reads “Similarly, genomes encoding just *nor* were denoted as non-denitrifiers (n=3,064)” (L473)

a. Perhaps due to the use of ‘could,’ it sounds like you could do something if you had the data (L149), but then the data wasn’t available (L153), but then proxies were used. If I replace ‘we could make a more direct assessment’ with ‘we attempted to make a more direct assessment’ then I think the progression of statements sounds less round-about. There might be other examples like this (e.g., on L163 ‘could calculate’ vs ‘calculated’).

Response: We have modified the language to state “We subsequently assessed the degree to which complete and partial denitrifiers were associated with high or low resource availability using CoverM³⁴ to map reads from the TARA Oceans project³⁵ to MAGs derived from the same data.” (L198). L163 (now L207) has been modified to read “We calculated a standardized environmental response”. We have checked for similar ways of expressing our findings and have revised accordingly.

b. I don't work with environmental data sets like this but using chlorophyll as a proxy seemed problematic to me. I would expect it to correlate with electron donors but perhaps correlate even more with O₂ levels. The problem I see is where dissolved O₂ is used as a proxy for NO₃⁻, assuming a negative correlation. Is interdependence between these parameters not an issue? Would it be easy to show data to illustrate that these are independent proxies?

Response: We think there was a misunderstanding: we used nitrate concentrations directly, not oxygen concentrations as a proxy for nitrate concentrations in our analysis. We have excluded the prior mentioning of the correlation between nitrate and oxygen to reduce confusion (previously L170) and have revised the text to clarify that chlorophyll was used as a proxy for carbon availability in the marine environments (L203-206). The text reads "Nitrate was used as a proxy for denitrification electron acceptor availability and chlorophyll concentration as a proxy for electron donor supply". Note that TARA uses a UV sensor for nitrate, which has lower sensitivity for nitrite compared to nitrate. Thus, the measurement of nitrate + nitrite largely represents nitrate.

The reviewer is correct that oxygen and chlorophyll are overall correlated in the dataset, albeit to a lesser degree than oxygen and nitrate ($\rho = 0.38$ vs. 0.53). Since we mapped reads on an ecoregion-by-ecoregion basis to enhance the chance that the MAGs represented the denitrification genotype found in a sample, we did not have enough replicates to statistically isolate the effects oxygen compared to chlorophyll and nitrate. Therefore, we decided to focus on chlorophyll and nitrate as the variables directly related to our hypothesis that electron donors and denitrification electron acceptor availability drives the prevalence of complete and partial denitrifiers.

c. I wonder if Fig 3 should be deprioritized, or even omitted, given the above and considering that the correlations were weak compared to those with the environmental parameters in Figs 5 and 6.

Response: We agree, and figure 3 has been moved to the supplement and is now Supplementary Figure 3.

3. I recommend giving another proofread with a focus on how some sentences might be rephrased to facilitate understanding. I give a few specific examples below but there are more sentences that need attention.

Response: We have carefully read the final version of the manuscript and hope the text reads well after the revision.

4. It might help to comment on what assumptions are being made when interpreting the data. For example, partial denitrifiers might be less versatile/flexible with respect to nitrogen oxides but we are blind to what other metabolic capabilities are available for each organism (e.g., other respiratory pathways, fermentation, phototrophy; L257).

Response: We have substantially expanded the description of genomic differences between partial and complete denitrifiers. We are now presenting information on other metabolic

capabilities (L165-196 and in a new figure, now Figure 2) and highlight that complete denitrifiers are flexible in multiple dimensions, including broader range of organic substrates, higher transcription factor and transporter density, and a higher probability of encoding the ability to use other terminal electron acceptors and inorganic electron donors compared to partial denitrifiers.

Other comments:

5. The sequential order of complete vs various subgroups of partial denitrifiers should be made consistent between figures.

Response: The order of the denitrifier groups has been made consistent between the figures.

6. Similarly, try to describe panels in the order they appear (e.g., L235).

Response: The panel and text order now coincide.

7. L184 vs L187 Maybe better to use only 'difference' instead of introducing 'delta.'

Response: We understand the point of simplifying, but decided to keep the delta in the text, as we use this terminology in the figures, and it is not otherwise defined until the methods.

8. L227 – 'up to' vs 'above'

Response: we believe that the term "up to" is accurate in this context, while "above" describes the opposite of the observed pattern, but we have added the word "increases" to further clarify: "Increases in soil organic carbon content up to ~1% positively affected $\delta nos-nir$, while the effect of soil NO_3^- content was always negative" (L277).

9. L231 – I don't see NO_2^- concentrations shown

Response: The studies reported the sum of NO_3^- and NO_2^- and not the concentration of the individual nitrogen species. We have modified the text to make that clear and in line with the axis label in figures 4 and 5. As mentioned above, TARA uses a UV sensor, which has lower sensitivity for nitrite compared to nitrate. Thus, the measurement largely represents nitrate.

10. L232 – confusing sentences describing a 'dramatic shift in prediction.' Maybe describe the trend more explicitly and give the temperature data its own sentence.

Response: The sentence now reads (L281-285) "Within the marine dataset, predicted $\delta nos-nir$ was lower at the highest concentrations of the sum of NO_3^- and NO_2^- compared to the lowest concentrations, and showed a dramatic shift in prediction from a decreasing to increasing relationship with temperature at 22 °C (Fig. 4b)."

11. L258 – how is stability of growth rate inferred?

Response: The stability of the growth rate was inferred based on the fact that organisms with lower maximum growth rates cannot have the same range of growth rates as organisms with high maximum growth rates. L 363 now reads “while partial denitrifiers are restricted to slower and therefore less variable growth rates.”

12. Fig 5 and 6 – I didn’t see an explanation for the shaded squares along the x-axes.

Response: These are tick marks indicating data points, so the denser they are, the more values we have for a given value of a variable. For example, in figure 5a, we have a lot of datapoints with potassium between 0 and 400 mg kg⁻¹, but fewer between 500 and 900 mg kg⁻¹. This has been added to the figure captions “Vertical marks on the x axis in each panel denote data density.”

Reviewer #2 (Remarks to the Author):

This study presents a meta-analysis of curated genomic data from isolates and ecosystems to co-locate known genes (using HMMs) involved in the steps of denitrification to determine the prevalence of partial versus complete pathways and the ecosystem-based variables that drive these trends. While the analyses here are a good step forward towards demonstrating that partial denitrifiers tend to dominate over complete denitrifiers in ecosystems, a trend that has been alluded to in other studies, there are some flaws and biases in the modeling exercise that limit the overall impact of the conclusions.

Response: We have revised the manuscript and included additional analyses to address the limitations mentioned in the specific comments by the reviewer.

The analysis relies completely on in silico tools to determine prevalent types of denitrifiers within their niches based on how fast they grow or how they access resources. Thus, the data represent a model of microbial behavior and thus include the biases of algorithms and sampling schemes (e.g. heavy reliance on cultivated strain genomes). For instance, the gRodon tool used to predict growth rate is based on a presumed direct correlation between codon bias and growth rates in complete genomes; hence, this tool is less predictive of growth rates for the MAGs and SCGs relative to the complete isolate genomes. As such, the conclusions of this work present averages calculated across diverse types of data and do not necessarily reflect ecosystem or phylotype diversity in terms of ecosystem denitrifying potential.

Response: We thank the reviewer for their concern about using gRodon. While it is true that gRodon is based on a correlation between codon usage bias (CUB) and growth rate in isolates, it is not based on a direct correlation. Rather, there is a correlation between CUB metrics and maximum growth rates in fast growing organisms, but CUB saturates in the genomes of slow growing taxa. Following the gRodon guidelines, we divided genomes into “fast” and “slow” growers to differentiate between the growth rates correlated and uncorrelated with CUB. Furthermore, with respect to the completeness of genomes, gRodon was shown to be primarily biased when there are few (i.e. fewer than 10) ribosomal

proteins used for the annotation. Therefore, we excluded genomes with fewer than 10 ribosomal proteins from our analysis. Nonetheless, even if we just include the isolate genomes (corresponds to 49% of our dataset), initiators, terminators and initiator-terminators are still inferred to less-often be fast-growing than complete denitrifiers (p-value of χ^2 test < 0.003 in all cases). Similarly, looking just at MAGs (51% of our dataset), all groups of partial denitrifiers are inferred to have a lower frequency of fast-growers compared to complete denitrifiers (47-49% genomes with fast growth in partial denitrifiers compared to 63% in complete denitrifiers; $P = 3.11 \times 10^{-16}$). Therefore, we are confident that our statement that complete denitrifiers are more likely to grow faster than partial denitrifiers holds. Additionally, there is evidence that CUB is correlated with growth-rate *in-situ* based on mapping reads to MAGs (ex. metagenome mapping in aquatic samples (Long et al <https://doi.org/10.1038/s41396-020-00773-1>), and qSIP in soil samples during rewetting (Chuckran et al. <https://doi.org/10.1073/pnas.241303212>). This provides an additional line of evidence that CUB can also be a reliable metric for the growth rates of MAGs.

We agree with the reviewer that our study looks at global patterns of denitrifier types among genomes rather than genomes from specific biomes. However, we do not view this as a weakness. Combining MAGs and isolates and sampling bacterial diversity as evenly as possible by taking representatives from GTDB allows us to observe how widely distributed the complete denitrification trait is, independent of the environment and in line with our objective to survey the commonness of the complete denitrification trait. Our added analyses (next comment) add further characterization of traits distinguishing complete and partial denitrifiers. On the other hand, looking at the imbalance between the initiation and termination of denitrification in environmental metagenomes allows us to infer how “complete” or balanced denitrification is at the community level, and comparing the prevalence of clade I and II *nosZ* provides insight into the prevalence of partial and complete denitrifiers among nitrous oxide reducers at the organism level, all independent of the cultivability or ease of assembly of their genomes. We observed wide variation in gene ratios between samples within some marine compared to soil environments, indicating that our results do indeed capture differences in ecosystem denitrifying potential. Thus, we believe our global analyses provide valuable information about the distribution of denitrifier types among bacteria and in the environment, and what characterizes these denitrifier types, and the environmental drivers of their distribution.

One nagging question is how much would growth rate depend on metabolic flexibility, for instance facultative oxygen respiration in an ecosystem context? If there is a correlation between growth rate and complete denitrification, there are likely other physiological factors at play that are not purely related to denitrification. The model presented here has some gaps that lead to misconceptions about how denitrification and partial versus complete pathways are distributed based on niche. The model should be refined to include other factors that are known about complete versus partial denitrifiers, namely their ability to use oxygen, organic carbon, other key regulated pathways (e.g. switches between N₂O respiration and DNRA) in populations, and whether the genes come from lithotrophs (e.g. *nirK* in AOA) that would bias the data towards a partial denitrification phenotype. For instance, it is known that AOA can not denitrify but nearly all

encode nirK and AOB encode nirK and norB genes. In other words, how much does metabolic diversity of the encoding microbial populations skew the generalized niche preference model presented in this study of partial and complete denitrifiers? Can this metabolic life-style diversity be accounted for (or discounted) somehow in the models presented here under "denitrification" when denitrifying genes are utilized by microorganisms in different metabolic contexts?

Response: The reviewer poses many good questions here, which we have addressed by 1) adding an extensive analysis of other genome traits associated with complete vs. partial denitrification and 2) clarifying why a possible bias towards partial denitrifiers as indicated by the reviewer is not of concern in our analysis.

The new analysis we added includes ability to metabolize and grow on different substrates (organic acids, amino acids, sugars), oxidize and reduce different inorganic substrates, and the number of transporters and transcription factors encoded by the selected genomes. We expanded the second paragraph in the introduction to provide background on this new analysis (L54-75) and have revised the Results (L. 165-196 and new Figure 2), Discussion (L. 331, 360, 339) and Methods (L. 480-516) sections.

For this substantial revision of the manuscript, we switched out the database of denitrifier genomes to provide a more up to date representation of the full known diversity of bacteria found in the environment (GTDB v214.1). With this we also expanded the number of genomes analyzed (previously 48,284, now 61,293). Compared to our previous database, these genomes had been quality controlled and dereplicated using identical criteria, and align our results with the scientific community's increasing tendency to use GTDB as a starting point for comparative and ecological genomics studies (ex. <https://doi.org/10.1038/s41467-023-43435-4>). With the exchange to the bacterial GTDB genomes we are not including archaea from the comparative genomics part of the work but have confirmed this does not affect our overall prevalence about the dominance of partial denitrifiers (complete denitrifiers account for 22.9% of genomes encoding *nir* and/or *nosZ* when we include Archaea, compared to 22.8% when we exclude them). This is mainly because archaea encoding Nir fall almost exclusively in Nitrososphaerota, which we already excluded in the original analysis, and also because archaea are much rarer in genome databases than bacteria. Thus, the overall patterns reported in our previous submission hold, namely partial denitrifiers dominating over complete denitrifiers, faster growth among complete denitrifiers, and complete denitrifiers being substantially more prevalent among organisms encoding clade I compared to clade II NosZ.

We understand the concern about a possible bias towards partial denitrification caused by AOA. However, the dominance of partial denitrifiers cannot be attributed to overrepresentation of AOA and/or anammox bacteria since both AOA and anammox bacteria were excluded in the original version of the manuscript, as their nitrite reductases are either known not to be or not known to be involved in respiratory denitrification. We also recognize it may not have been clearly described in the original manuscript and have clarified this in the revised manuscript (L. 475-478 for genomes and 599-603 for metagenomes). The AOB are included since they are known to denitrify. However, there are only a few AOB genomes and

based on our previous nitrite reductase-centered analysis of the same metagenomes, we can say that AOB make up a very small fraction of nitrite reducers (Pold et al. ISME Communications 2024). Overall, we conclude that the analyses in both the original and revised manuscripts are not biased due to inclusion of nirK from AOA since these were omitted from the analysis.

Could these models be tested and validated with gene expression datasets? It is not difficult to run the same HMMs on transcriptomes, and this could deconvolute the process differences (e.g. nitrification from denitrification)

Response: As suggested, we have added gene expression datasets in the revised version of the manuscript. We agree that evaluating the expression patterns of nitrite and nitrous oxide reductases in metatranscriptomes to evaluate if they are actually expressed adds valuable information to our study. We have repeated the GraftM analysis using a selection of 413 metatranscriptomes, and report the results on L297-315 and in Fig. 6 in the revised version. Overall, these results are consistent with those from the comparative genomics and metagenomics, indicating a dominance of *nir* over *nosZ* expression and of clade II over clade I *nosZ* expression, in addition to relatively lower $\delta nos-nir$ expression under conditions associated with low carbon inputs and/or high nitrate availability.

line 73- For the introduction, it might be helpful to include a small figure showing the steps in the denitrification pathway to explain the definitions of different types (e.g. initiators, terminators, complete, and partial) as not all readers will be familiar with these pathways/processes.

Response: Thank you for the suggestion. The definitions of the steps of the denitrification pathway are now shown in the revised version of figure 1b and its caption, and we refer to this figure when describing our framework of initiators, terminators and complete denitrifiers (L. 90).

line 164 - It doesn't seem surprising that denitrification would correlate with nitrate supply, since this is its primary substrate. This is essentially a confirmatory finding.

Response: The reviewer is correct that this is mainly a confirmatory finding, but also a validation of the analysis. We have removed this sentence in the revised version and the figure is now in the supplement.

line 171 - if the genomes in the dataset are not representative of prevalent groups, then does that negate some of the averaged findings reported above? The lack of relevant, comprehensive, sampling and heavy reliance on cultivated strain genomes seems like a major problem for this study that aims to find a comprehensive model for denitrifier diversity.

Response: We are not heavily relying on cultivated isolates and with the new dataset of genomes we used, approximately half of the genomes (51%) consist of MAGs and single-cell genomes derived from the environment. Ideally, we should have included more MAGs or only used MAGs but the practicality is that MAGs are almost uniformly of lower quality compared to isolate genomes. Analyzing the presence and absence of specific genes and

their co-occurrences, as in our study, is particularly sensitive to low quality genomes if the genes occur in the parts of genomes poorly captured in MAGs. By restricting the analysis to high-quality assemblies, we had to discard many available MAGs. We have added text in the methods (L457-460) to state the tradeoff between accuracy and being representative when selecting genomes. For the comparative genomics part, our selection of genomes is the best of what is currently available, but as long read sequencing of environmental samples increases the quality of associated MAGs will increase. The sentence referred to has been deleted during the revision of the manuscript, but we highlight the need for high quality genomes of representative environmental genomes in the discussion (L. 433-436).

line 195 - the hypothesis that nitrate assimilation has a higher rate than denitrification is testable as the authors could look at the ratio of nitrate assimilatory genes relative to denitrification genes.

Response: We appreciate the reviewer's comment but annotating nitrate assimilatory genes using short reads relative to denitrification genes is not straightforward. A new reference tree and GraftM database would need to be made and manually curated for each assimilatory protein prior to searching the metagenomes. Furthermore, the purpose of this sentence was not to say decisively that there is greater nitrate assimilation, but rather discuss based on the literature that this could be a possibility.

line 209 - what is the rationale for why sewage communities for *nosZ* would be different than all of the other environments?

Response: It is not clear why clade I is so favored in the sewage metagenomes, but considering sewage samples were collected from all over the world with varying inputs of industrial, human, and agricultural wastes, it seems to be a legitimate pattern. Metadata from these samples is restricted to sociological and antibiotic factors rather than biogeochemical factors, which limits our ability to draw conclusions about the relevance of our hypothesized drivers of complete and partial denitrifier prevalence. Clade I *nosZ* reads in these samples are dominated by *nosZ* sequences associated *Acidovorax*, a genus commonly found in wastewater treatment plants. We have now added this into the text on L258-260.

line 226 - if less than half of the variance in gene ratios was explained by the models, then shouldn't the models be better refined?

Response: The models explained less than half of the variance except for δ *nosZI-nosZII* in soils where 59% were explained. The main issue is the limited metadata associated with the metagenomes. The variables included in the final model followed a rigorous selection process, which ensured that we had the best model given the data available. First, we excluded the highly correlated variables. Then we used 100 iterations of VSURF, a random forest-based variable selection process that excluded variables with no importance for the prediction of gene ratios, selected that subset of variables that are strongly related to the

gene ratios, and then selected predictor variables that optimized the prediction. We only included variables kept by the model in at least 95% of the iterations in our final model. We have added additional information into the methods to help clarify how we selected the variables in the final models (L672-674).

line 297 - why couldn't these additional drivers (e.g. organic matter) be included in the present study?

Response: Organic matter quality data was not collected as metadata in the metagenome studies we have analyzed. This sentence has been deleted during the revision of the manuscript since the discussion has been reorganized. However, in the analyses of metatranscriptomes that were added during the revision, we included studies where soils were amended with substrates known to include carbon compounds readily-metabolized by soil microorganisms (e.g. root exudates, glucose or glycine addition), and here we see that carbon addition drove the balance towards termination over initiation of denitrification transcripts (L. 304-308).

Reviewer #1 (Remarks to the Author):

The authors have done a commendable job addressing reviewer comments. I have one lingering comment.

The new analysis of metabolic breadth was a welcome addition. However I was surprised by the choice of acceptors, which I would have expected to be phylogenetically constrained (though I could be wrong), while others were omitted:

Answer: We thank the reviewer for their comment. The reviewer is correct that individual terminal electron acceptor and donor associated enzymes are likely to be restricted to specific areas of the organism phylogeny. However, we analyzed counts and phylogenetic signal in raw gene presence/absence does not necessarily imply that model residuals will show phylogenetic autocorrelation. Further, phylogenetic autocorrelation is primarily problematic if the residuals are more clustered than random (Brownian motion) on the phylogeny because this indicates pseudoreplication. We did not find the residuals to be more clustered than expected under Brownian motion. Nonetheless, we are aware that Brownian motion may not be the best null model to use for bacterial denitrification, where multiple evolutionary events play a substantial role in the phylogenetic distribution of the enzymes (Jones et al. 2008. doi:10.1093/molbev/msn146). However, we are unaware of a better way to handle this.

a. Why wasn't oxygen considered as a potential electron acceptor in Fig. 2e. I expect that aerobic respiration might be common and could correlate with other findings.

Answer: We originally checked for oxygen as terminal electron acceptor by searching for terminal oxidases (cytochrome *c* oxidases and cytochrome *bd* oxidase) but were not 100% confident in our annotations and therefore excluded them. We did not identify some of the cyt-*bd* proteins in our original search that are reported in Murali et al. (doi: 10.1038/s41396-021-01019-4), which we now realize may be due to an incomplete original HMM. We have now revisited the analysis and by using lithogenie (<https://github.com/Arkadiy-Garber/MagicLamp>), we have found what we believe to be cyt-*bd* in genomes where Murali et al. also searched but did not identify the enzyme. We also originally thought to exclude oxygen as electron acceptor because most denitrifiers are expected to be facultative aerobes. With the inclusion of the terminal oxidases in the revised manuscript, we show that oxygen is near-uniformly used by denitrifiers (99% of genomes in our analysis, with 85% encoding at least one low affinity and 92% encoding at least one high affinity enzyme). Because these numbers are higher than any other electron acceptor, we have not added them to Fig. 2e as it would distort the figure. However, we have included the new information on oxygen as terminal electron acceptor in the text as follows:

Methods L508-513: “Genomic potential for aerobic respiration was identified by screening the HMM-based HCO family search used to annotate Nor for HCO classes A, B and C, and completing a second HMM-based search using the cytochrome *bd* oxidase family reference alignment from Murali et al. The proteins were categorized

into high (HCO classes B and C and cytochrome *bd* oxidase) and low affinity oxidases (HCO class A) following Morris and Schmidt.”

Results L185-194: “Although most genomes lacked the genetic potential to oxidize inorganic substances or reduce non-nitrogenous terminal electron acceptors (median = 0) **other than oxygen**, complete denitrifiers had the genetic potential to both oxidize and reduce more compounds than partial denitrifiers (Kolmogorov-Smirnov statistic $P < 0.0001$ in all cases). **Almost all (99%) genomes encoded the potential for aerobic respiration, with low affinity oxidases more prevalent in complete denitrifiers (87%) than initiators (83%; $X^2=15.03$, $P = 0.0001$), and similar in prevalence to terminators (85%; $X^2=0.083$, $P = 0.7734$) and initiator-terminators (86%; $X^2=0.91$, $P = 0.3397$). High affinity variants were more prevalent among complete denitrifiers (99%) vs. initiators (89%; $X^2=228$, $P < 2.2 \times 10^{-16}$), terminators (94%; $X^2=84.63$, $P < 2.2 \times 10^{-16}$) and initiator-terminators (88%; $X^2=142.82$, $P < 2.2 \times 10^{-16}$).**”

Discussion L378: “Under carbon-rich conditions supporting rapid growth in the environment, complete denitrifiers may benefit from using available $\text{NO}_3^-/\text{NO}_2^-$ more efficiently ²¹, while under rapidly fluctuating conditions many have the flexibility to use whichever electron donor and terminal electron acceptors are most readily available, **including oxygen via high affinity terminal oxidases.**”

b. The authors might also consider DMSO and TMAO, which might be important in marine environments. However, I understand that this is also getting away from the original intent of the manuscript.

Answer: We agree with the reviewer’s point that there are organic terminal electron acceptors that may be important in certain biomes. We did not include these because there are many organic compounds that can be reduced (humic compounds, DMSO, TMAO, fumarate, nitrobenzenes), so we focused on inorganic ones (except for reductive halogenases) to simplify. The intention was not to have an exhaustive list, but rather a broad selection to address if complete denitrifiers are more metabolically flexible.

However, for the reviewer’s interest, we have extracted the DMSO reductases from HMSS2 (assuming a minimum subunit composition of DmsABC for the membrane-bound enzyme and DorCA for the periplasmic enzyme). These results confirm our overall observation that complete denitrifiers are more likely to encode the genetic potential to use other terminal electron acceptors, with 17% of complete denitrifiers vs. 6% of initiators, 4% of initiator-terminators, and 2% of terminators encoding DMSO reductases. However, this information is not included in the revised manuscript.

c. If you end up looking for the above acceptors, I don’t suppose that fumarate reductase (for fumarate respiration) can be reliably distinguished from succinate dehydrogenase?

Answer: While we are not experts in these two enzymes, we believe that the reviewer is correct that fumarate reductase cannot be readily distinguished from succinate

dehydrogenase because the enzymes form polyphyletic and intermingled clades (ex.[https://doi.org/10.1016/S0005-2728\(01\)00239-0](https://doi.org/10.1016/S0005-2728(01)00239-0)). Furthermore, in some bacteria these enzymes are bifunctional, completing both reactions. And as mentioned above, the intention was not to have an exhaustive list, and we decided to focus on a range of inorganic electron acceptors.

Reviewer #2 (Remarks to the Author):

The authors have meticulously responded to each of the reviewer comments and added additional data that clarifies and better validates their findings. In going through the revision, I do not see any major concerns that require comment.

Answer: Thank you to the reviewer for their kind feedback.